# On Flow-based Generative Models for Probabilistic Forecasting

## Abstract

Flow-based generative models (FBGM) have emerged as a dominant approach to generative modeling in many domains for their scalability and controllability, but have notably not made the same impact on autoregressive probabilistic forecasting. Although the methodology behind these models can be applied directly to the time series setting, and in theory offers the potential to apply the advances in generative modeling to time series, this direct approach is difficult to use in practice. In this work, we investigate this methodological gap by generalizing the key elements of flow-based generative modeling to the time series setting to devise a more practical related algorithm. We show that FBGMs based on linear stochastic differential equations are instances of a more general mean-field variational inference algorithm for conditional exponential family distributions that constructs Bayes estimators of natural parameters. This insight yields a family of mean-squared error based latent probabilistic forecasters that contains a discrete time counterpart of FBGMs for time series. We demonstrate that the models we develop inherit the convenient theoretical properties of FBGMs while being easy to work with in practice.

## 1 Introduction

Flow-based generative models (FBGM), including denoising diffusion, score based diffusion, and flow matching models, have become the dominant approach to generative modeling. These models represent a stochastic differential equation (SDE) that transforms samples from a known prior distribution into samples from an unknown target distribution, and often use a different recipe for solving the generative modeling problem compared to traditional approaches. This alternative approach is highly scalable [Ramesh et al., 2022, Podell et al., 2023, Saharia et al., 2022], can leverage conditioning information in flexible ways [Dhariwal and Nichol, 2021, Ho and Salimans, 2022], and can be controlled in order to incorporate user defined dynamics [Liu et al., 2024, Domingo-Enrich et al., 2024, Havens et al., 2025]. Furthermore, FBGMs are capable of learning from paired data. If $x_0$ and $x_1$ are samples from an unknown joint distribution $p(x_0, x_1)$, then one can use the same approach to construct an SDE whose transition distribution from $t = 0$ to $t = 1$ is $p(x_1|x_0)$ [De Bortoli et al., 2023]. Given this capability, it directly follows that this approach could, in principle, be used to construct an SDE to model time series data. If $p(x_{1:N}) = p(x_1) \prod_{k=1}^{N-1} p(x_{k+1}|x_{1:k})$ represents the unknown distribution of time series data, then each of the transition terms, $p(x_{k+1}|x_{1:k})$, can be interpreted as a target distribution for a FBGM in the paired data setting where the data pairs are consecutive elements of the time series, $(x_{k+1}, x_k)$, and the previous elements $x_{1:k-1}$ can be thought of as extra conditioning information. In theory, learning this kind of model for time series would inherit the scalability and controllability that FBGMs possess, allowing practitioners to port over the recent advances in generative modeling to time series applications. However, this approach has surprisingly only recently been explored [Chen et al., 2024a, Tamir et al., 2024, Park et al., 2024, Chen et al., 2024b] even though diffusion based time series models have been studied for several

years [Yang et al., 2024, Meijer and Chen, 2024]. We attribute this gap to the practical numerical difficulties associated with training and sampling from these models as one must first learn, and then simulate, a stochastic differential equation, with potentially non-smooth dynamics, over a long time domain compared to the short time domain encountered in standard generative modeling. To address this problem, we develop a discrete time version of Neural SDEs derived from FBGMs that are founded on the same theoretical principles, while being substantially easier to work with in practice. We do this by generalizing two key elements needed to construct FBGMs, stochastic interpolation and the Markovian projection, to the time series setting, where they become Gaussian condition random fields and a form of mean-field variational inference respectively. We construct a family of latent probabilistic time series models that are closely related to existing time series models, including MSE based non-probabilistic forecasters and conditional Gaussian autoregressive models, and compare their performance on various latent probabilistic forecasting problems.

## 2  Background

We will first review how flow-based generative models are constructed and then build intuition for how to go about generalizing this construction to the time series setting. Suppose that $p(y_0, y_1)$ is a joint distribution over a source and target random variable. The (paired) generative modeling problem is to find a parametric approximation of $p(y_1|y_0)$ [1]. Flow-based generative models solve this problem by constructing, and then learning, a *latent* SDE whose transition distribution from times $t = 0$ to $t = 1$ is $p(y_1|y_0)$. There are three steps involved in constructing and learning this SDE - **stochastic interpolation**, the **Markovian projection**, and **matching**.

**Stochastic interpolation** [Albergo and Vanden-Eijnden, 2023] is used to interpolate between probability distributions by defining interpolations between their samples. For example, consider the joint distribution $p(x_0, x_t, x_1)$, where $x_t = (1 - t)x_0 + tx_1$ and $(x_0, x_1) \sim p(x_0, x_1)$. By the definition of $x_t$, it is true that $p(x_{t=1}) = p(x_1)$, and also that $p(x_{t=1}|x_0) = p(x_1|x_0)$, so we verify that the marginal distribution of $x_t$ interpolates between $p(x_0)$ and $p(x_1)$. In practice, one assumes that at times $t = 0$ and $t = 1$, $x_0 := y_0$ and $x_1 := y_1$ so that $p(x_t)$ is an interpolation between $p(y_0)$ and $p(y_1)$.

A popular method for constructing stochastic interpolants, which we use in this paper, is conditioning a user-defined base SDE, whose diffusion coefficient does not depend on the current state, to start at $x_0$ and end at $x_1$. This SDE takes the form $dx_t = b_t(x_t)dt + L_t dW_t$ where $b_t(x_t)$ is the drift of this base SDE and $L_t$ is the diffusion coefficient. This SDE is used to construct a joint distribution of the form $p(x_0, x_t, x_1) = p(x_t|x_0, x_1)p(x_0, x_1)$ where $p(x_t|x_0, x_1)$ is the probability of $x_t$ when the base SDE has been conditioned to start at $x_0$ and end at $x_1$. In order to solve the generative modeling problem of $p(x_1|x_0)$, FBGMs are constructed as an SDE whose marginal distribution is $p(x_t|x_0)$. This is accomplished using the **Markovian projection**.

**Proposition 1** (Markovian projection SDE [Shi et al., 2024]). *Let $p(x_1|x_0)$ be a conditional distribution over target variables given source variables and let $p(x_t|x_0, x_1)$ denote the distribution of the base SDE $dx_t = b_t(x_t)dt + L_t dW_t$ when conditioned to start at $x_0$ and end at $x_1$. The "Markovian projection SDE" is an SDE whose marginal distribution, denoted by $q^*(x_t|x_0)$ is equal to $p(x_t|x_0)$. It is given by:*

$$dx_t = (b_t(x_t) + L_t L_t^T \mathbb{E}_{p(x_1|x_0,x_t)} \left[ \nabla \log p(x_1|x_0, x_t) \right]) dt + L_t dW_t \tag{1}$$

See Prop 3. of [De Bortoli et al., 2023] for a proof. Proposition 1 is a solution to the paired generative modeling problem because $q^*(x_{t=1}|x_0) = p(x_1|x_0) := p(y_1|y_0)$. Given a sample from the source distribution, $x_0 \sim p(x_0)$, we can simulate the SDE from $t = 0$ to $t = 1$ to generate a sample from the target distribution. However, this SDE contains an intractable drift term that depends on the posterior distribution of $x_1$ given $x_0$ and $x_t$. This is addressed using a **matching** learning objective. For example, in score matching, [Vincent, 2011, Song et al., 2021], one writes the drift in the following variational form:

$$\nabla \log q^*(x_t|x_0) = \operatorname*{argmin}_{s_t(x_t, x_0)} \mathbb{E}_{p(x_0, x_1, x_t)} \left[ \left\| L_t L_t^T \nabla \log p(x_1|x_0, x_t) - s_t(x_t, x_0) \right\|^2 \right] \tag{2}$$

---

[1] The unpaired setting is when we do not condition on $y_0$.

If $s(x_t, x_0; \theta)$ is parameterized by a neural network, then one can minimize this expectation using the standard machine learning toolkit to find the Markovian projection SDE. However, obtaining a Monte Carlo estimate of the expectation for stochastic gradient descent requires being able to sample from $p(x_0, x_1, x_t)$, which requires simulation of the base SDE. As such, the base SDE is chosen so that this distribution is tractable. After training is complete, then the flow-based generative model is given by the SDE $dx_t = (b_t(x_t) + L_t L_t^T s_t(x_t, x_0))dt + L_t dW_t$. In general, matching algorithms, such as score matching, drift matching and bridge matching, are algorithms for learning the Bayes estimator of a random variable because of the well known relationship between posterior expectations and mean squared error [Jaynes, 2003]:

**Proposition 2** (Bayes estimate of parameter). *Let $p(z, \theta)$ be a joint distribution and let $\theta^*(z)$ be the Bayes estimate of $\theta$ based on $z$ under the squared error risk. Then the Bayes estimate takes the following two forms:*

$$\theta^*(z) = \mathbb{E}_{p(\theta|z)}[\theta] = \underset{f(z)}{\operatorname{argmin}} \ \mathbb{E}_{p(z,\theta)} \left[ \|f(z) - \theta\|^2 \right] \tag{3}$$

See Appendix C.3 for a derivation. In score matching, one would have $z = (x_0, x_t)$ and $\theta = \nabla \log p(x_1|x_0, x_t)$, while other matching approaches, such as flow matching [Albergo and Vanden-Eijnden, 2023, Lipman et al., 2023, Liu et al., 2023] and bridge matching [Shi et al., 2024].

Given the strong theoretical, interpretability, and empirical results of FBGMs, one might expect that a direct application to time series would inherit the same benefits. However, this approach has surprisingly only recently been explored [Chen et al., 2024a,b, Tamir et al., 2024, Park et al., 2024] even though diffusion based time series models have been studied in a different manner for several years [Yang et al., 2024, Meijer and Chen, 2024]. We attribute this gap to the challenges that the time series setting presents to flow-based methods compared to settings such as image generation. In the standard image generation setting, there is no coupling between the prior and data distributions, and so one can learn SDEs that can be easily simulated with a few number of function evaluations [Liu et al., 2023, Pooladian et al., 2023]. However, SDEs that are constructed to model time series data present a challenge during inference due to compounding numerical errors that are attributed to either a mismatch between the learned model and data, or due to the numerical solver itself, get accumulated during generation which can lead to poor performance in practice. Discrete time autoregressive models, on the other hand, do not suffer from these issues to the extent that Neural SDEs do and are much more widely used in practice. With this in mind, we aim to understand find a discrete time version of FBGMs for time series that will work better in practice.

## 3 Method

We present a generalization of the FBGM construction for the time series setting.

### 3.1 Generalized linear stochastic interpolation

Recall that stochastic interpolation constructs a distribution over a latent stochastic process, which we denote by $\mathbf{x}$, that is sampled from a base SDE that is conditioned to start at $x_0 := y_0$ and end at $x_1 := y_1$. Our generalization of stochastic interpolation is founded on the observation that many of the base SDEs used in practice are linear SDEs, and that the FBGM recipe is unchanged if we introduce Gaussian potential functions to relax the endpoint conditions. Since linear SDEs have Gaussian transition distributions, they can naturally be combined with these Gaussian potentials to construct a Gaussian conditional random field. This conditional random field will serve as our tool for stochastic interpolation, which we call "generalized linear stochastic interpolation".

Let $y_{\tau_{1:T}}$ denote time series data that is generated by an unknown distribution $p(y_{\tau_{1:T}})$. For brevity, we assume that $\tau_{1:T}$ is the same for all time series, but note that our theory accommodates datasets with series sampled at different times. We will construct, and perform inference, in the distribution $p(\mathbf{x}|y_{\tau_{1:T}})$, which we will obtain by conditioning a linear SDE on user defined Gaussian potential functions. The potential function at time $t_k \in \mathcal{R}$ will be denoted by $\phi(x_{t_k}|\theta_{t_k}(y_{\tau_{1:T}}))$, where $\theta_{t_k}$ the the natural parameter of the Gaussian that arbitrarily depends on $y_{\tau_{1:T}}$. See Appendix C for a review of exponential family distributions. We also use the notation $\phi_{k+1|k}(x_{k+1}|x_k) = N(x_{k+1}|Ax_k + u, \Sigma)$ to denote a Gaussian transition distribution from $x_k$ to $x_{k+1}$ with state transition matrix $A$, bias vector $u$ and covariance matrix $\Sigma$.

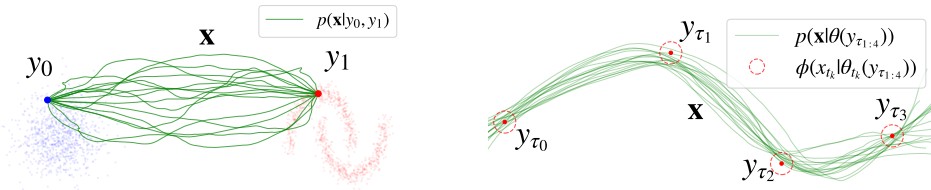

(a) Stochastic interpolation  (b) Generalized stochastic interpolation

Figure 1: Generalized stochastic interpolation incorporates Gaussian potential functions to relax the endpoint conditions of stochastic interpolation and is applied to time series data.

### 3.1.1 Gaussian conditional random fields

Chain structured Gaussian CRFs are a tractable class of probabilistic models that are widely used in time series modeling (CITE):

**Definition 1** (Conditional Random Field [Lafferty et al., 2001, Sutton et al., 2012])**.** *Let $x_{1:N}$ be a sequence of random variables, $\phi_{k+1|k}(x_{k+1}|x_k)$ be a set of Gaussian transition distributions between consecutive variables, and $\phi(x_k|\theta_k)$ a set of Gaussian potential functions with natural parameters $\theta_k \in \theta$. A conditional random field (CRF) is a probability distribution given by:*

$$p(x_{1:N}|\theta) \propto \prod_{k=1}^{N-1} \phi_{k+1|k}(x_{k+1}|x_k) \prod_{k=1}^{N} \phi(x_k|\theta_k) \quad (4)$$

Due to the chain-structure of $p(x_{1:N}|\theta)$ and the fact it is jointly Gaussian, inference can be performed efficiently using message passing. The backward messages, defined below, will play a significant role in our theory:

**Proposition 3** (Backward messages)**.** *The $k$'th backward message associated with the CRF in Definition 1 is defined with the following recurrence relation:*

$$\phi(x_{k-1}|\beta_{k-1}) = \int \phi_{k|k-1}(x_k|x_{k-1})\phi(x_k|\theta_k + \beta_k)dx_k, \quad \beta_N = 0 \quad (5)$$

*where $\theta_{k+1} + \beta_{k+1}$ denotes the direct sum of $\theta_{k+1}$ and $\beta_{k+1}$. This recurrence also uniquely identifies a function, denoted by $\Phi_{k,k+1}$ that performs the parameter updates as:*

$$\beta_k = \Phi_{k,k+1}(\theta_{k+1} + \beta_{k+1}) \quad (6)$$

Note that each $\beta_k$ is a function of $\theta_{k+1:N}$. See Appendix D for a full derivation of sequential and parallel message passing, and Appendix H for pseudo code and implementation considerations. Although we do not focus on the forward messages, they are defined with analogous recurrence relations to the backward messages and can be used to extend our methodology to flow-matching models for time series forecasting (see Corollary 5). CRFs offer an efficient way to model the latent variables at a fixed set of times, but are not immediately suited for continuous time.

### 3.1.2 Linear time-invariant stochastic differential equations

We will use linear-time invariant SDEs to construct the transition distributions of continuous time CRFs. Linear time-invariant SDEs (LTI-SDEs) are SDEs of the form $dx_t = Fx_tdt + LdW_t$, where the drift matrix $F$ and diffusion coefficient matrix $L$ are constant with respect to $t$ and $x_t$. LTI-SDEs have the convenient property that their transition distribution is available in closed form [Särkkä and Solin, 2019, Singhal et al., 2023]. The transition distribution from $x_t$ to $x_{t+s}$, where $s > 0$ is an increment of time, is given by

$$\phi_{t+s|t}(x_{t+s}|x_t) = N(x_{t+s}|A_sx_t, \Sigma_s), \quad \text{where} \begin{bmatrix} A_s & \Sigma_s A_s^{-T} \\ 0 & A_s^{-T} \end{bmatrix} := \exp \left\{ \begin{bmatrix} F & LL^T \\ 0 & -F^T \end{bmatrix} s \right\} \quad (7)$$

We use LTI-SDEs for their tractability, but note that our theory is completely compatible with more general linear SDEs. One can directly plug in this transition distribution into a CRF in Definition 1 to obtain a conditional random field over a continuous time domain. However, we can be more general. In the next proposition, we highlight a relationship between conditioned linear SDEs and CRFs ([Särkkä et al., 2006, Särkkä and Solin, 2019]):

**Proposition 4** (Conditioned LTI-SDE). *Let $\phi_{t+s|t}(x_{t+s}|x_t)$ be the transition distribution of the LTI-SDE $dx_t = Fx_t dt + LdW_t$ and let $\{\phi(x_{t_k}|\theta_{t_k})\}_{t_k \in \mathcal{R}}$ be potential functions at times in the set $\mathcal{R}$. Then the piecewise-linear SDE,*

$$dx_t = (Fx_t + LL^T \nabla \log \phi(x_t|\beta_t))dt + LdW_t, \quad x_{t_1} \sim \phi(x_{t_1}|\beta_1 + \theta_1) \tag{8}$$

*where $t \in (t_k, t_{k+1})$ and $t_k, t_{k+1} \in \mathcal{R}$, has a joint distribution at the times $t_{1:N} = \mathcal{T} \supseteq \mathcal{R}$ that is given by a CRF:*

$$p(x_{t_{1:N}}|\theta) \propto \prod_{t_k \in \mathcal{T}} \phi_{t_{k+1}|t_k}(x_{t_{k+1}}|x_{t_k}) \prod_{t_k \in \mathcal{R}} \phi(x_{t_k}|\theta_{t_k}) \tag{9}$$

*where $\beta_t = \Phi_{t,t_{k+1}}(\theta_{t_{k+1}} + \beta_{t_{k+1}})$.*

See appendix Appendix E.1 for the full proof and Corollary 5 for a nice expression for the associated probability flow ODE in terms of both the forward and backward messages. Proposition 4 suggests that a practical way to work with conditioned linear SDEs in practice is convert them into CRFs on a discretization of the time domain so that inference can be performed via message passing. This results in the ability to sample and perform inference in linear SDEs $O(\log |\mathcal{T}|)$ time on parallel compute [Hassan et al., 2021, Corenflos et al., 2021, Smith et al., 2023]. The conditioned SDE Proposition 4 is our main tool for stochastic interpolation as it gives us the ability to sample from $p(\mathbf{x}|\theta(y_{\tau_{1:T}}))$ at an arbitrary discretization of the time domain.

## 3.2 Target probabilistic model for FBGM

Recall that in the FBGM recipe, we used the stochastic interpolation to construct a joint distribution over the interpolant and the data, $p(y_0, x_t, y_1)$, before performing the Markovian projection. We can take the same step here to construct a joint distribution over $y_{\tau_{1:T}}$ and $\mathbf{x}$ using the data distribution, $p(y_{\tau_{1:T}})$ and the distribution of the interpolant, $p(\mathbf{x}|y_{\tau_{1:T}}) := p(\mathbf{x}|\theta(y_{\tau_{1:T}}))$.

**Definition 2** (Target joint distribution). *Let $p(y_{\tau_{1:T}})$ be the distribution of observed time series data and let $p(\mathbf{x}|y_{\tau_{1:T}})$ be the distribution of the generalized linear stochastic interpolant, which is the distribution of a linear SDE conditioned on the user defined potential functions $\{\theta_{t_k}(y_{\tau_{1:T}})\}_{t_k \in \mathcal{R}}$ at the times $\mathcal{R}$, as in Proposition 4. Then the induced joint distribution over $\mathbf{x}$ at the times $t_{1:N} = \mathcal{T} \supset \mathcal{R}$ and $y_{\tau_{1:T}}$ is given by:*

$$p(x_{t_{1:N}}, y_{\tau_{1:T}}) = p(y_{\tau_{1:T}}) \left( \frac{1}{Z(y_{\tau_{1:T}})} \prod_{t_k \in \mathcal{T}} \phi_{t_{k+1}|t_k}(x_{t_{k+1}}|x_{t_k}) \prod_{t_k \in \mathcal{R}} \phi(x_{t_k}|\theta_{t_k}(y_{\tau_{1:T}})) \right) \tag{10}$$

*where $Z(y_{\tau_{1:T}})$ is the partition function of $p(x_{t_{1:N}}|y_{\tau_{1:T}})$.*

Before continuing, it is crucial that we understand this joint distribution and the role it plays in the FBGM recipe. Unlike the standard approach to generative modeling where one defines a joint distribution by defining a prior over the latent variable and a likelihood distribution over the data, the FBGM uses an alternate construction to build $p(\mathbf{x}, y_{\tau_{1:T}})$ using the data distribution directly. Furthermore, the tools FBGMs employ are fundamentally designed for probabilistic inference in $\mathbf{x}$ instead of $y_{\tau_{1:T}}$. Since $\mathbf{x}$ is completely user designed through the choice of base LTI-SDE and potential functions, we are able to solve a wide range time series problems.

Suppose we split each sequence of data into observed and unobserved portions, $y_{\tau_{1:T}} = (y_{\mathcal{O}}, y_{\mathcal{U}})$, where $y_{\mathcal{O}}$ is a subsequence that we observe at both train and test time while $y_{\mathcal{U}}$ is only observed at training time, as is the case in time series forecasting.[2] The ability to perform inference in $p(\mathbf{x}|y_{\mathcal{O}})$ would solve a general latent probabilistic forecasting problem that reduces to the standard forecasting problem if the Gaussian potential functions are chosen as dirac delta functions -

---

[2]This also covers the imputation setting, but we do not explore this in the interest of keeping a narrow scope.

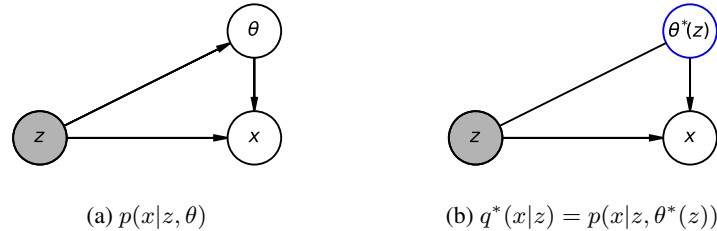

(a) $p(x|z,\theta)$  (b) $q^*(x|z) = p(x|z,\theta^*(z))$

Figure 2: The CMFVI approximation of $p(x|z)$ is $q^*(x|z)$. Choosing $(x, z, \theta) = (x_{t_{1:N}}, y_{\mathcal{O}}, \theta(y_{\tau_{1:T}}))$ recovers $q^{\text{MSE}}$, $(x, z, \theta) = (x_{t_k}, (x_{t_{1:k-1}}, y_{\mathcal{O}}), \theta(y_{\tau_{1:T}}))$ recovers $q^{\text{MSE-AR}}$ and $(x, z, \theta) = \lim_{s \to 0}(x_{t+s}, (x_t, x_{t_{1:k-1}}, y_{\mathcal{O}}), \theta(y_{\tau_{1:T}}))$ for $t \in (t_k, t_{k+1})$ recovers $q^{\text{Neural-SDE}}$.

$\phi(x_{t_k}|\theta_{t_k}(y_{\tau_{1:T}})) := \delta(x_{t_k} - y_{t_k})$. For example, if one chooses the LTI-SDE to be the Wiener velocity model [Särkkä and Solin, 2019, Särkkä et al., 2006] and potential functions of the form $\phi(x_{t_k}|\theta(y_{\tau_{1:T}})) \propto N(x_{t_k}|y_{t_k}, \sigma^2 I)$, then inference in $p(\mathbf{x}|y_{\mathcal{O}})$ corresponds to forecasting the smoothed position and velocity of the particle whose positions were observed at $y_{\tau_{1:T}}$. However, $p(\mathbf{x}|y_{\mathcal{O}})$ is intractable because $p(y_{\tau_{1:T}})$ is arbitrary. To this end, we develop variational inference algorithms for this task.

## 3.3 Neural latent SDE for latent probabilistic forecasting

The first inference algorithm we develop is a direct extension of flow-based generative models to the latent probabilistic forecasting setting. For a fixed discretization of the time domain, we can treat consecutive latent variables $(x_{t_k}, x_{t_{k+1}})$ as elements of a paired dataset with the previous elements $x_{t_{1:k-1}}$ and observations $y_{\mathcal{O}}$ as extra conditioning information. This lets us directly apply the existing FBGM recipe to construct a conditional, piecewise SDE to solve the latent probabilistic forecasting problem.

**Proposition 5** (Neural latent SDE). *Let $p(x_{t_{1:N}}, y_{\tau_{1:T}})$ be the joint distribution defined in Definition 2 and suppose that $y_{\tau_{1:T}} = (y_{\mathcal{O}}, y_{\mathcal{U}})$, where $\mathcal{O}$ and $\mathcal{U}$ are the times at which sequences are observed and unobserved at test time, respectively. Then the neural latent SDE is the following piecewise SDE:*

$$dx_t = (F_t x_t + L_t L_t^T \nabla \log \phi(x_t | \beta_t^*(x_t, x_{t_{1:k}}, y_{\mathcal{O}})))dt + L_t dW_t, \tag{11}$$

$$\text{where } \beta_t^*(x_t, x_{t_{1:k}}, y_{\mathcal{O}}) = \mathbb{E}_{p(y_{\mathcal{U}}|x_t, x_{t_{1:k}}, y_{\mathcal{O}})}[\beta_t(y_{\tau_{1:T}})], \text{ and } t \in (t_k, t_{k+1}) \tag{12}$$

*Furthermore, the transition distribution of this SDE from time $t_k$ to $t_{k+1}$ is $p(x_{t_{k+1}}|x_{t_{1:k}}, y_{\mathcal{O}})$. We will use $q^{\text{Neural-SDE}}$ to denote the path measure associated to this SDE.*

See Appendix G.2 for a proof and Appendix G for the general constructions of the score function, Markovian projection SDE and probability flow ODE. By construction, Proposition 5 can be used to solve the latent probabilistic forecasting problem because it has the correct joint distribution over the latent space. Furthermore, its form is almost identical to that of its base LTI-SDE in Proposition 4, except that its parameter, $\beta^*$, is the Bayes estimator of a backward message. We will show next that models of this form can be derived by solving a constrained mean-field variational inference problem.

## 3.4 Constrained mean-field variational inference

Next we introduce our main contribution which is the variational inference algorithm underlying FBGMs, which we call "constrained mean-field variational inference". Given a conditional exponential family distribution $p(x|z,\theta)$, CMFVI constructs a variational approximation of $p(x|z)$ that is given by $p(x|z,\theta^*(z))$ where $\theta^*(z)$ is the Bayes estimator of $\theta$ given $z$. We first introduce CMFVI in an abstract way and then show how it can be used to do variational inference on the latent probabilistic forecasting distribution, $p(x_{t_{1:N}}|y_{\mathcal{O}})$.

Suppose that $z$ is a random variable, $\theta \sim p(\theta|z)$ is the natural parameter of an exponential family distribution, and $x \sim p(x|z,\theta)$ is a random variable drawn from a conditional exponential family of the form $p(x|z,\theta) = \exp\{\langle t_z(x), \theta \rangle - A(z,\theta)\}$. For intuition, assume that $x$ represents the future of a stochastic process, $z$ represents its past , and $\theta$ represents the parameters of this process. Furthermore,

suppose that the parameters are only available at training time so that at test time, sampling $x$ given $z$ requires the ability to sample from $p(x|z)$. Our goal is to predict the future of the process given its past, which requires the ability to sample from $p(x|z)$, however this distribution is intractable because $p(\theta|z)$ is arbitrary. To this end, we introduce a variational approximation of $p(x|z)$ using an algorithm closely resembling mean field variational inference, which we call "constrained mean field variational inference" (CMFVI):

**Theorem 1** (Constrained mean field VI solution). *Let* $p(x|z,\theta) \propto \exp\{\langle t_z(x), \theta \rangle - A(z,\theta)\}$ *be a conditional exponential family distribution with* $\theta \sim p(\theta|z)$. *The constrained mean field VI approximation of* $p(x|z)$, *denoted by* $q^*(x|z)$, *is defined as follows:*

$$q^*(x|z) = \underset{q(x|z)}{\operatorname{argmin}} \operatorname{KL}\left[q(x|z)p(\theta|z)\|p(x,\theta|z)\right] \tag{13}$$

$$= p(x|z, \theta^*(z)), \quad where \ \theta^*(z) = \mathbb{E}_{p(\theta|z)}[\theta] \tag{14}$$

See Appendix F.1 for a proof, Lemma 4 for equivalent expressions for the objective involving $\operatorname{KL}[q^*(x|z)\|p(x|z)]$ and a term resembling the mutual information between $x$ and $\theta$ given $z$. The parameter $\theta^*(z)$ is the Bayes estimator of $\theta$ given $z$ and by Proposition 2 can be learned using mean squared error minimization, provided that it is possible to sample from $p(z,\theta)$. While this variational approximation is tractable, it seems restrictive because it is a conditional random field and only exact when $\theta$ and $x$ are conditionally independent given $z$. However, this may not be a terrible assumption in the time series setting. If the process is deterministic, then we should be able to compute $x$ directly from $z$ without needing to know $\theta$, and so this independence assumption will hold because one will be able to compute the future values of the process directly from its past. In fact, in Corollary 8, we show that a direct application of CMFVI to $p(x_{t_{1:N}}|y_{\mathcal{O}})$, by selecting $x = x_{t_{1:N}}$, $z = y_{\mathcal{O}}$ and $\theta = \theta(y_{\tau_{1:T}})$, exactly recovers MSE based non-probabilistic forecasters, which are clearly capable of learning deterministic processes (see Corollary 8). We denote the model in Corollary 8 by $q^{\text{MSE}}$. In general, provided that the process is not too stochastic, we might expect that given a long enough history and a short enough prediction horizon that CMFVI could yield a reasonable approximation of $p(x|z)$, and perhaps with an infinitely short prediction horizon we may recover something exactly. This intuition motivates the use of CMFVI for learning the autoregressive factors of $p(x_{t_{1:N}}|y_{\mathcal{O}})$ in order to construct an autoregressive model to solve the probabilistic forecasting problem.

Suppose that $p(x_{t_k}|x_{t_{1:k-1}}, y_{\mathcal{O}})$ is one of the autoregressive factors of the latent forecasting distribution $p(x_{t_{1:N}}|y_{\mathcal{O}})$. We can use CMFVI to approximate each of the $k$ factors by setting $x = x_{t_k}$, $z = (x_{t_{1:k-1}}, y_{\mathcal{O}})$ and $\theta = \theta(y_{\tau_{1:T}})$:

**Proposition 6** (CMFVI transition approximation). *Let* $p(x_{t_{1:N}}|y_{\mathcal{O}})$ *be the target distribution and consider its* $k$'th *autoregressive factor* $p(x_{t_k}|x_{t_{1:k-1}}, y_{\mathcal{O}})$. *Then the CMFVI transition approximation is given by:*

$$q^{\text{transition}}(x_{t_k}|x_{t_{1:k-1}}, y_{\mathcal{O}}) \propto \phi_{t_k|t_{k-1}}(x_{t_k}|x_{t_{k-1}})\phi(x_{t_k}|\beta_{t_k}^*(x_{t_{1:k-1}}, y_{\mathcal{O}})) \tag{15}$$

*where* $\beta_{t_k}^*(x_{t_{1:k-1}}, y_{\mathcal{O}}) = \mathbb{E}_{p(y_{\mathcal{U}}|x_{t_{1:k-1}}, y_{\mathcal{O}})}[\beta_{t_k}(y_{\tau_{1:T}})]$ *is the Bayes estimate of* $\beta_{t_k}(y_{\tau_{1:T}})$, *which is defined using the message passing update operator* $\Phi_{t_k, t_{k+1}}$ *from Definition 7 as:*

$$\beta_{t_k} = \begin{cases} \Phi_{t_k, t_{k+1}}(\beta_{t_{k+1}}(y_{\tau_{1:T}}) + \theta_{t_{k+1}}(y_{\tau_{1:T}})) & \text{if } t_{k+1} \in \mathcal{R} \\ \Phi_{t_k, t_{k+1}}(\beta_{t_{k+1}}(y_{\tau_{1:T}})) & \text{otherwise} \end{cases} \tag{16}$$

See Proposition 6 for a proof. The form of Proposition 6 almost exactly matches the transition distribution of $p(x_{t_{1:N}}|y_{\tau_{1:T}})$ in Proposition 12, except that the backward messages are replaced with their Bayes estimators. We will use $q^{\text{transition}}$ to construct an autoregressive approximation model that will be a discrete time version of the Markovian projection SDE.

To use CMFVI to construct a discrete time version of FBGMs for time series, we will need to make the assumption that the covariances of the potential functions are independent of the values of $y_{\tau_{1:T}}$. This assumption holds in both the data space forecasting setting where we use dirac delta potential functions, and also in the case where the CRF is constructed as a linear dynamical system with constant observation noise. In this setting, it is also possible to rewrite $q^{\text{Neural SDE}}$ in a more interpretable form where the only unknown value is the mean of the next backward message:

**Corollary 1** (Neural latent SDE using potentials with fixed covariances). *If the covariance matrices associated with* $q^{\text{Neural SDE}}$ *are constant with respect to* $y$, *then the SDE associated with* $q^{\text{Neural SDE}}$ *is:*

$$dx_t = (F_t x_t + L_t L_t^T \nabla \log N(x_t | \mu_t^{\beta^*}(x_t, x_{t_{1:k-1}}, y_{\mathcal{O}}), \Sigma_t^{\beta}))dt + L_t dW_t \tag{17}$$

where $t \in (t_{k-1}, t_k)$, $\Sigma_t^\beta$ is the covariance of $\phi(x_t | \beta_t(y_{\tau_{1:T}}))$ and $\mu_t^*(x_t, x_{t_{1:k-1}}, y_{\mathcal{O}})$ is the Bayes estimator for it's mean.

The result follows directly from converting $\beta_{t_k}$ from natural parameters to standard parameters of a Gaussian and the linear equivariance of the Bayes estimator Appendix F.2. Note that by our assumption that the parameters of the potential functions do not depend on $y_{\tau_{1:T}}$, $\Sigma_t^\beta$ can be computed by performing message passing on $p(x_{t_{1:N}} | \varnothing_{\tau_{1:T}})$, where $\varnothing_{\tau_{1:T}}$ is an empty (or random) sequence sampled at the same times as $y_{\tau_{1:T}}$.

## 3.5 Discrete time Markovian projection

We propose an conditional Gaussian autoregressive model whose transition distributions are given by $q^{\text{transition}}$, which we denote by $q^{\text{MSE-AR}}$. We will directly relate it to Markovian projection SDE $q^{\text{Neural-SDE}}$ by associating $q^{\text{MSE-AR}}$ with a piecewise linear SDE that closely resembles $q^{\text{Neural-SDE}}$.

**Proposition 7** (Autoregressive CMFVI solution). *Let $p(x_{t_{1:N}} | y_{\mathcal{O}})$ be the target distribution, assume that the covariance matrices of its potential functions are constant with respect to $y$. The autoregressive model whose transitions are CMFVI solution, denoted by $q^{MSE-AR}$ is given by:*

$$q^{MSE-AR}(x_{t_{1:N}} | y_{\mathcal{O}}) \propto p(x_{t_1} | y_{\mathcal{O}}) \prod_{t_k \in \mathcal{T}} \phi_{t_k | t_{k-1}}(x_{t_k} | x_{t_{k-1}}) N(x_{t_k} | \mu_{t_k}^{\beta^*}(x_{t_{1:k-1}}, y_{\mathcal{O}}), \Sigma_{t_k}^\beta) \quad (18)$$

*where $\Sigma_{t_k}^\beta$ and $\mu_{t_k}^{\beta^*}(x_{t_{1:k-1}}, y_{\mathcal{O}})$ are the same as in Corollary 1. Furthermore, $q^{MSE-AR}$ has the same joint distribution over $x_{t_{1:N}}$ as the following piecewise linear SDE:*

$$dx_t = (F_t x_t + L_t L_t^T \nabla \log N(x_t | \mu_t^{\beta^*}(x_{t_{1:k-1}}, y_{\mathcal{O}}), \Sigma_t^\beta)) dt + L_t dW_t, \quad x_{t_1} \sim p(x_{t_1} | y_{\mathcal{O}}) \quad (19)$$

*where $\mu_t^*(x_{t_{1:k-1}}, y_{\mathcal{O}})$ is the Bayes estimator for the mean of $\beta_t(y_{\tau_{1:T}}) = \Phi_{t, t_k}(\beta_{t_{k+1}}(y_{\tau_{1:T}}))$, $\Sigma_t^\beta$ is its covariance matrix and $t \in (t_{k-1}, t_k)$ for $k = 2, \dots, T$.*

See Appendix F.3 and Definition 9 for a proof. A comparison of the piecewise linear SDE associated with $q^{\text{MSE-AR}}$ with the piecewise SDE associated to $q^{\text{Neural-SDE}}$ reveals why we interpret $q^{\text{MSE-AR}}$ as the discrete time version of the Markovian projection SDE. We can see that the only difference between the two SDEs are their Bayes estimators for $\mu_t^\beta(y_{\tau_{1:T}})$:

$$q^{\text{MSE-AR}} : \mu_t^{\beta^*}(x_{t_{1:k}}, y_{\mathcal{O}}) = \mathbb{E}_{p(y_{\mathcal{U}} | x_{t_{1:k}}, y_{\mathcal{O}})} \left[ \mu_t^\beta(y_{\tau_{1:T}}) \right]$$

$$q^{\text{Neural-SDE}} : \mu_t^{\beta^*}(x_t, x_{t_{1:k}}, y_{\mathcal{O}}) = \mathbb{E}_{p(y_{\mathcal{U}} | x_t, x_{t_{1:k}}, y_{\mathcal{O}})} \left[ \mu_t^\beta(y_{\tau_{1:T}}) \right]$$

The only difference between the two Bayes estimators is their dependence on the current state $x_t$. If $x_t$ does not carry more information about $y_{\mathcal{U}}$ compared to what is already available from $x_{t_{1:k}}$ and $y_{\mathcal{O}}$, then we can expect that $q^{\text{MSE-AR}}$ and $q^{\text{Neural-SDE}}$ will model nearly the same distribution. As we will show in our experiments, this is something that one can expect in the time series setting because data is usually sampled frequently enough where the extra capacity that $q^{\text{Neural-SDE}}$ has over $q^{\text{MSE-AR}}$ may not make enough of an impact in practice to warrant using $q^{\text{Neural-SDE}}$ in practice. We introduced three different CMFVI based time series models - $q^{\text{MSE}}$ 8, $q^{\text{MSE-AR}}$ 7 and $q^{\text{Neural-SDE}}$ 1 which use CMFVI to joint distribution, transition distributions, and infinitesimal transitions of the target distribution respecitvely. All of these models are Gaussian, and are therefore closely related to existing time series models.

## 3.6 Connection to traditional time series models

The CMFI-based time series models that we have developed all have an autoregressive Gaussian structure which makes them related to existing time series models. First, when one chooses potential functions to align with the data times $\mathcal{R} = \tau_{1:T}$, then $q^{\text{MSE}}$ is identical to MSE based non-probabilistic forecasters, which are are trained to predict the future of a time series, $y_{\mathcal{U}}$ given an observed history, $y_{\mathcal{O}}$. Next, $q^{\text{MSE-AR}}$ is a conditional Gaussian autoregressive model that is trained to minimize a mean-squared error based objective. This model is in the same family as conditional Gaussian models that are trained for maximum likelihood, but differ in that $q^{\text{MSE-AR}}$ can be though of parameterizing the mean of each transition distribution whereas maximum likelihood models parameterize both the mean and covariance. Overall, the models that we have developed can be seen as mean-squared error based time series models for probabilistic forecasting where the uncertainty in the models only depend on the time in between observations and not the observations themselves.

|  | Brusselator | Double Pendulum | FitzHugh | Lorenz | Lotka | Van der Pol |
|---|---|---|---|---|---|---|
| MSE | $3.04 \pm 0.69$ | $9.03 \pm 0.34$ | $27.75 \pm 4.50$ | $5.91 \pm 0.60$ | $2.16 \pm 1.18$ | $-0.77 \pm 0.01$ |
| AR-MSE | $0.49 \pm 0.18$ | $0.61 \pm 0.02$ | $15.08 \pm 1.18$ | $8.82 \pm 0.29$ | $0.12 \pm 0.25$ | $-0.59 \pm 0.01$ |
| AR-MLE (Latent) | $3.39 \pm 1.91$ | $0.43 \pm 0.01$ | $13.10 \pm 2.48$ | $8.49 \pm 1.05$ | $0.23 \pm 0.27$ | $-0.70 \pm 0.00$ |
| AR-MLE (Obs.) | $3.79 \pm 2.05$ | $0.42 \pm 0.01$ | $13.35 \pm 2.47$ | $7.77 \pm 0.76$ | $0.11 \pm 0.32$ | $-0.70 \pm 0.00$ |
| FBGM (Latent) | $2.06 \pm 1.12$ | $0.56 \pm 0.03$ | $6.15 \pm 0.75$ | $12.11 \pm 0.80$ | $0.17 \pm 0.42$ | $-0.69 \pm 0.00$ |
| FBGM (Obs.) | $0.93 \pm 0.29$ | $0.51 \pm 0.01$ | $11.67 \pm 1.80$ | $5.28 \pm 0.50$ | $0.47 \pm 0.67$ | $-0.71 \pm 0.00$ |

(a) Negative log likelihood (lower is better)

|  | Brusselator | Double Pendulum | FitzHugh | Lorenz | Lotka | Van der Pol |
|---|---|---|---|---|---|---|
| MSE | $0.56 \pm 0.02$ | $0.99 \pm 0.00$ | $2.15 \pm 0.16$ | $1.09 \pm 0.01$ | $0.50 \pm 0.02$ | $0.48 \pm 0.00$ |
| AR-MSE | $0.59 \pm 0.01$ | $1.16 \pm 0.01$ | $3.58 \pm 0.27$ | $1.25 \pm 0.01$ | $0.55 \pm 0.03$ | $0.52 \pm 0.00$ |
| AR-MLE (Latent) | $0.65 \pm 0.04$ | $1.27 \pm 0.01$ | $2.32 \pm 0.17$ | $1.26 \pm 0.03$ | $0.59 \pm 0.03$ | $0.52 \pm 0.01$ |
| AR-MLE (Obs.) | $0.66 \pm 0.05$ | $1.27 \pm 0.01$ | $2.37 \pm 0.13$ | $1.26 \pm 0.04$ | $0.58 \pm 0.03$ | $0.52 \pm 0.01$ |
| FBGM (Latent) | $0.62 \pm 0.05$ | $1.20 \pm 0.01$ | $2.34 \pm 0.17$ | $1.09 \pm 0.03$ | $0.55 \pm 0.03$ | $0.49 \pm 0.01$ |
| FBGM (Obs.) | $0.64 \pm 0.02$ | $1.17 \pm 0.01$ | $2.29 \pm 0.15$ | $1.08 \pm 0.02$ | $0.55 \pm 0.03$ | $0.51 \pm 0.00$ |

(b) Normalized root mean squared error (lower is better)

Table 1: Evaluation metrics for our models (MSE and AR-MSE) for probabilistic forecasting compared to baseline models trained in both the latent and data spaces.

## 4 Experiments

We compare the performance of our models versus other approaches to time series modeling in latent probabilistic forecasting on dynamical system datasets. We created 6 synthetic datasets representing noisy observations of dynamical systems. Our models used a Wiener velocity model as our base SDE and emission potentials of the form $\phi(x_{t_k}|\theta_{t_k}(y_{\tau_{1:N}})) \propto N(y_{t_k}|x_{t_k}, \sigma^2 I)$. Our models, $q^{\text{MSE}}$ and $q^{\text{MSE-AR}}$, and the baseline models were trained to approximate the probabilistic forecasting distribution $p(x_{t_{k+1:N}}|x_{t_{1:k}}, y_{\mathcal{O}})$. See Appendix I for details about the datasets, parameters used for stochastic interpolation and other implementation details. Our models, $q^{\text{MSE}}$ and $q^{\text{MSE-AR}}$, were each trained using mean squared error to learn their respective Bayes estimators. We used a non-autoregressive FBGM trained with flow-matching and a conditional Gaussian chain trained for maximum likelihood as our baselines. We trained each of these baselines in two ways to learn $p(x_{t_{k+1:N}}|x_{t_{1:k}}, y_{\mathcal{O}})$. First, we trained these baseline models to learn the latent distribution directly by learning directly from samples from $p(x_{t_{1:N}}|y_{\tau_{1:N}})$. Second, we trained these models in the observation space to learn $p(y_{\mathcal{U}}|y_{\mathcal{O}})$ directly, and at test time, produced latent samples $x_{t_{k+1:N}}$ by first sampling $y_{\mathcal{U}}$ using $y_{\mathcal{O}}$, and then sampling from the stochastic interpolator using the full sequence $(y_{\mathcal{O}}, y_{\mathcal{U}})$. For all of the autoregressive models, instead of learning the distribution of the first point $p(x_{t_{k+1}}|y_{\mathcal{O}})$, we produced a heuristic sample by sampling from the stochastic interpolant that is only conditioned on $y_{\mathcal{O}}$. We always chose $t_{k+1}$ to be a time contained in $\mathcal{O}$ in order for this heuristic to give reasonable samples. For each model, we trained using 5 different seeds and report the (empirical) negative log likelihood and normalized root mean squared error of samples from the true distribution, $p(x_{t_{k+1:N}}|y_{\mathcal{U}})$, using 32 sampled trajectories from each model, averaged over each dimension and time step. In all of our models, we used a one layer recurrent neural network with a GRU cell as we found that this model had sufficient model capacity to represent our data. Our results are displayed in Table 1. We can see that the AR

## 5 Conclusion

We showed how to generalize the elements that comprise flow-based generative models to the time series setting and uncovered a discrete time version of these models that shares convenient properties that FBGMs possess, including a closed form solution and Bayes estimator parameters. Our framework also encapsulates other existing time series models, including MSE based non-probabilistic forecasters and conditional Gaussian autoregressive models. This unified perspective sheds light into the role that FBGMs can play in time series.

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

## A Appendix

The appendix contains proofs and implementation details for the main paper. It is organized as follows:

1. Related work Appendix B
2. Background Appendix C
   - Exponential family distributions Appendix C.1
   - Mean field variational inference Appendix C.2
   - Bayes estimation Appendix C.3
3. Message passing (D)
   - Sequential message passing (D.1)
   - Parallel message passing (D.2)
   - Basic probabilistic queries (D.4)
4. Conditioned linear SDEs (E)
   - Conditioned linear SDEs (E.1)
   - Basic probabilistic queries (E.2)
   - Corresponding probability flow ODE (E.3)
5. Constrained mean field VI (F)
   - Derivation (F.1)
   - Bayes estimator equivariance (F.2)
   - CMFVI time series models (F.3)
6. Flow-based generative models (G)
   - Score function of FBGMs (G.1)
   - General form of Markovian projection SDE (G.2)
   - General form of Markovian projection ODE (G.3)
7. Message passing implementation details (H)
   - Numerical stability considerations (H.1)
   - Message passing pseudocode (H.2)
8. Dataset details (I)
9. Model implementation details (J)

## B Related Work

There are numerous perspectives on flow-based generative models [Luo, 2022, Dieleman, 2023] and even more variants of these models. At their core, these models start by constructing a stochastic process that starts at a prior distribution and ends at the data distribution. Diffusion models use progressive noising of data to build this map [Sohl-Dickstein et al., 2015, Ho et al., 2020, Song et al., 2021] via a simple SDE whose stationary distribution is Gaussian. On the other hand, flow-matching models [Liu et al., 2023, Albergo and Vanden-Eijnden, 2023, Lipman et al., 2023] use a stochastic bridge to build this map by conditioning a simple SDE to start at a point in the prior distribution and end at the data distribution. The choice of simple SDE used in all of these models is a user-defined choice that typically is a linear SDE, such as variance preserving SDE [Song et al., 2021], Brownian motion, Ornstein-Uhlenbeck process, and others, due to their tractability as Gaussian processes [Särkkä and Solin, 2019], and is even used to construct more exotic latent SDEs such as critically damped langevin dynamics [Dockhorn et al., 2022, Chen et al., 2024c] or the Weiner velocity model [Bar-Shalom et al., 2001, Särkkä et al., 2006]. In our paper, we abstract away these choices and generally consider using linear SDEs to construct the initial map between distributions. There are a few different ways to go from this initial stochastic process to a FBGM. A common way to construct a FBGM from this is construct and optimize and ELBO for the likelihood of data under this initial process [Kingma et al., 2021]. Alternatively, one can directly solve for the SDE whose marignal distribution is that of this initial process [Song et al., 2021, Lipman et al., 2023] or define it as the

SDE whose path measure is as close as possible to the initial process [Shi et al., 2024, De Bortoli et al., 2023] in terms of KL divergence, called the Markovian projection. We adopt the latter view over the ELBO view because it explicitly constructs a solution to the generative modeling problem and is available in closed form while this is hidden in the ELBO formulation and show that the solution to a mean field variational inference problem can be seen as an approximate discrete time counterpart.

Flow-based generative models have been successfully applied to time series problems in a *non-autoregressive* fashion [Kollovieh et al., 2023, Yuan and Qiao, 2024, Kollovieh et al., 2025, Hu et al., 2024, Yang et al., 2024, Meijer and Chen, 2024]. These models transform the time series generative modeling problem into the standard generative modeling problem used in image generation by treating each time series as a single vector by concatenating all times together, and then learning a map from a Gaussian vector of the same size to the data vector. These approaches can be conditioned using guidance [Rasul et al., 2021, Dhariwal and Nichol, 2021, Ho and Salimans, 2022, Kollovieh et al., 2023] which allows them to perform tasks such as forecasting and imputation. Our approach differs from these in that we construct autoregressive models.

The class of models most relevant to our paper are autoregressive neural SDEs that are trained using principles from flow-based generative models. [Chen et al., 2024a] uses a Főllmer process to model the transition distributions of the distribution of time series data, which is the same approach that we adopt in our Neural SDE model. [Park et al., 2024] also learns a similar latent Neural SDE model that uses a similar form of soft conditioning as us (through the use of emission potentials), and is trained to maximize the likelihood of data. [Tamir et al., 2024] is also similar where they perform stochastic interpolation using Gaussian processes and perform inference with Kalman smoothing as well, which is a form of message passing. Finally, [Shen and Cheng, 2025] learns a more general SDE to learn the distribution of time series data where the diffusion coefficient is not independent of the current state and also maximize the likelihood of data. These related papers are all related to the Neural SDE that we describe in our paper. Our main contributions are centered around investigating how to apply the approach used to construct these continuous time models for creating similar discrete time models. [El-Gazzar and van Gerven, 2025] used flow matching to learn the next state distribution of time series data, but did not learn a Főllmer process for this task and instead learned to transform a Gaussian into the next state distribution.

## C  Background

### C.1  Exponential family distributions

Our findings can be most easily written using exponential family distributions. Although we restrict our attention to Gaussian distributions, the form of our results are most readable in natural parameter space.

**Definition 3** (Exponential family distribution). *An probability distribution is in the exponential family if its density function can be written in the following form:*

$$p(x|\theta) = \exp\{\langle t(x), \theta \rangle - A(\theta)\} \tag{20}$$

*where $t(x)$ is called the sufficient statistic, $\theta$ the natural parameter and $A(\theta)$ the partition function.*

The member of this family that we will use is the multivariate Gaussian distribution. A multivariate Gaussian with mean $\mu$ and covariance matrix $\Sigma$ has the sufficient statistic $t(x) = (x, xx^T)$ and natural parameters $\theta = (-\frac{1}{2}\Sigma^{-1}, \Sigma^{-1}\mu)$. In practice, it is more convenient to drop the $-\frac{1}{2}$ scaling term and work with the parameters $(J, h) = (-\Sigma^{-1}, \Sigma^{-1}\mu)$, where $J$ is the precision matrix of the distribution. While these are not exactly the natural parameters, we will refer to them as so. Throughout this paper, we will work with unnormalized Gaussian distributions, which we call "Gaussian potentials". We use the notation $\phi(x|\theta)$ to denote a Gaussian potential function over $x$ with natural parameters $\theta$. A convenient property of the natural parameter form is that the score function takes a simple form.

$$\nabla \log \phi(x|\theta) = Jx - h \tag{21}$$

Another Gaussian distribution that we will use extensively is the Gaussian transition distribution. We write $\phi_{k+1|k}(x_{k+1}|x_k) = N(x_{k+1}|Ax_k + u, \Sigma)$ to denote the Gaussian transition distribution from $x_k$ to $x_{k+1}$ with state transition matrix $A$, bias vector $u$ and covariance matrix $\Sigma$.

## C.2  Mean field variational inference

Mean field variational inference is an approximate inference algorithm for probabilistic models. It's main feature is that it's solution is available in a simple closed form expression. Let $p(x, \theta)$ be a joint distribution over $x$ and $\theta$. The mean field variational problem is to find distributions, $q_x(x)$ and $q_\theta(\theta)$ that minimize the KL divergence between $q_x(x)q_\theta(\theta)$ and $p(x, \theta)$.

**Proposition 8** (Mean field variational inference for CRFs). *Let $p(\theta)$ be a distribution over $\theta$, $p(x|\theta)$ be the CRF in Definition 1 and $p(x, \theta) = p(\theta)p(x|\theta)$ be the joint distribution over $x$ and $\theta$. Then the solutions to*

$$\operatorname*{argmin}_{q_x(x), q_\theta(\theta)} \mathrm{KL}\left[q_x(x)q_\theta(\theta)|p(x, \theta)\right] \tag{22}$$

*will satisfy:*

$$q_x(x) \propto \exp\{\mathbb{E}_{q_\theta(\theta)}\left[\log p(x|\theta)\right]\} \tag{23}$$

$$q_\theta(\theta) \propto \exp\{\mathbb{E}_{q_x(x)}\left[\log p(\theta|x)\right]\} \tag{24}$$

See [Beal, 2003] for a proof. Typical use cases of mean field VI use tractable classes of distributions for $p(\theta)$ and $p(x|\theta)$ so that one can perform EM style, alternating updates to obtain the optimal $q$ distributions [Beal, 2003, Johnson et al., 2014]. However, in our setting, we will use mean field VI differently. We will assume nothing about the form of $p(\theta)$, but will constrain the variational problem by fixing $q_\theta(\theta) = p(\theta)$.

## C.3  Bayes estimation

**Lemma 1** (Bayes estimate of parameter). *Let $p(z, \theta)$ be a joint distribution and let $\theta^*(z)$ be the Bayes estimate of $\theta$ based on $z$ under the squared error risk. Then the Bayes estimate takes the following two forms:*

$$\theta^*(z) = \mathbb{E}_{p(\theta|z)}[\theta] = \operatorname*{argmin}_{f(z)} \mathbb{E}_{p(z,\theta)}\left[\|f(z) - \theta\|^2\right] \tag{25}$$

*Proof.* Let $\mathcal{L}[f]$ be the loss function defined as follows:

$$\mathcal{L}[f] = \mathbb{E}_{p(z)}\left[\|f(z) - \theta^*(z)\|^2\right]$$

Clearly, the minimizer of $\mathcal{L}[f]$ is $\theta^*(z)$. With a bit of rearranging and using Bayes rule, we can rewrite $\mathcal{L}[f]$ as follows:

$$
\begin{aligned}
\mathcal{L}[f] &= \mathbb{E}_{p(z)}\left[\|f(z) - \theta^*(z)\|^2\right] \\
&= \mathbb{E}_{p(z)}\left[\|f(z)\|^2\right] - 2\mathbb{E}_{p(z)}\left[\langle f(z), \theta^*(z)\rangle\right] + \underbrace{\mathbb{E}_{p(z)}\left[\|\theta^*(z)\|^2\right]}_{\text{const. w.r.t. } f} \\
&= \mathbb{E}_{p(z,\theta)}\left[\|f(z)\|^2\right] - 2\mathbb{E}_{p(z)}\left[\langle f(z), \mathbb{E}_{p(\theta|z)}[\theta]\rangle\right] + \text{const.} \\
&= \mathbb{E}_{p(z,\theta)}\left[\|f(z)\|^2\right] - 2\mathbb{E}_{p(z,\theta)}\left[\langle f(z), \theta\rangle\right] + \text{const.} \\
&\quad \text{(complete the square)} \\
&= \mathbb{E}_{p(z,\theta)}\left[\|f(z) - \theta\|^2\right] - \underbrace{\mathbb{E}_{p(z,\theta)}\left[\|\theta\|^2\right]}_{\text{const. w.r.t. } f} + \text{const.}
\end{aligned}
$$

The minimizer of $\mathcal{L}[f]$ is unaffected by the constant terms, and so we have that $\theta^*(z) = \mathbb{E}_{p(\theta|z)}[\theta]$ is the solution to

$$\operatorname*{argmin}_{f(z)} \mathbb{E}_{p(z,\theta)}\left[\|\theta - f(z)\|^2\right]$$

$\square$

 # D   Message passing

In this section we will review message passing and identify the key operations that are needed to perform message passing updates. We defer the discussion of numerically stable implementations of these operations to Appendix H. First we'll identify the key operations that are needed to perform message passing updates for the backward messages and then show how these operations can be used to perform message passing updates for the forward messages.

At a high level, the sequential and parallel message passing algorithms are variable elimination algorithms that eliminate different variables of the chain structured graph. The sequential algorithms operates on individual nodes and begins at one of the ends of the chain and sequentially eliminate variable at the end of the chain, whereas the parallel algorithm operates on pairs of nodes and eliminates the middle variable of the pair. For example, a rough sketch of the sequential elimination process looks like $(0), 1, 2, 3, 4 \to (1), 2, 3, 4 \to (2), 3, 4 \to (3), 4 \to (4)$, where the parentheses indicate the current node that is being processed. On the other hand, the parallel algorithm looks like $(0, 1), 2, 3, 4 \to (0, 2), 3, 4 \to (0, 3), 4 \to (0, 4)$.

## D.1   Sequential message passing

The sequential message passing updates for the backward messages can be written using the following recurrence relation:

$$\phi(x_{k-1}|\beta_{k-1}) = \int \phi_{k|k-1}(x_k|x_{k-1})\phi(x_k|\theta_k)\phi(x_k|\beta_k)dx_k, \quad \beta_N = 0 \tag{26}$$

See Appendix H.3 for pseudocode. There are two operations on Gaussians that are needed to perform these updates. The first is a "multiply" operation that takes two potential functions and returns a new potential function, and the second is an "update" operation that absorbs a potential function into a transition function.

**Definition 4** (Multiply). *Let $\phi_1(x)$ and $\phi_2(x)$ be potential functions over the same variable. Then the "multiply" operation is defined as*

$$\phi_1(x)\phi_2(x) \mapsto \hat{\phi}(x) \tag{27}$$

When $\phi_1(x)$ and $\phi_2(x)$ are parameterized using natural parameters, then the multiply operation simply adds the natural parameters, i.e. if $\theta_1$ and $\theta_2$ are the natural parameters of $\phi_1(x)$ and $\phi_2(x)$, then $\phi_1(x|\theta_1)\phi_2(x|\theta_2) \mapsto \phi_1(x|\theta_1 + \theta_2)$. We used this property to write the sequential message passing updates for the backward messages **??**. We do note that when one uses a different parameterization, the multiply operation may look different. We will examples of this in Appendix H.

The second operation is the "update" operation, which absorbs a potential function into a transition function. This operation is what handles the integral in the recurrence relation.

**Definition 5** (Update). *Let $\phi(y|x)$ be a transition function and $\phi(y)$ be a potential function over the first variable. Then the "update" operation is defined as*

$$\phi(y)\phi_{y|x}(y|x) \mapsto \hat{\phi}_{y|x}(y|x)\hat{\phi}(x) \tag{28}$$

*where $\hat{\phi}_{y|x}(y|x)$ and $\hat{\phi}(x)$ are a new transition function and potential function, respectively.*

Essentially, the update operation performs a change of variables of the coupling of $x$ and $y$ on the LHS. Furthermore, when the terms of the LHS are Gaussian, then the terms of the RHS are also Gaussian. This allows us to perform the update operation in closed form (see Appendix H).

The multiply and update operations are sufficient to perform the sequential message passing updates for the backward messages. For example, the backward message passing updates can be written as:

$$\int \phi_{k|k-1}(x_k|x_{k-1}) \underbrace{\phi(x_k|\theta_k)\phi(x_k|\beta_k)}_{\text{multiply} \to \phi(x_k|\theta_k+\beta_k)} dx_k \tag{29}$$

$$= \int \underbrace{\phi(x_k|\theta_k + \beta_k)\phi_{k|k-1}(x_k|x_{k-1})}_{\text{update} \to \hat{\phi}_{k|k-1}(x_k|x_{k-1})\phi(x_{k-1}|\beta_{k-1})} dx_k \tag{30}$$

$$= \underbrace{\int \hat{\phi}_{k|k-1}(x_k|x_{k-1})dx_k}_{\text{transition integrates to 1}} \phi(x_{k-1}|\beta_{k-1}) \tag{31}$$

$$= \phi(x_{k-1}|\beta_{k-1}) \tag{32}$$

The forward messages can be computed in a similar manner. The forward messages are given by:

$$\phi(x_{k+1}|\alpha_{k+1}) = \int \phi_{k+1|k}(x_{k+1}|x_k)\phi(x_k|\theta_k)\phi(x_k|\alpha_k)dx_k, \quad \alpha_1 = 0 \tag{33}$$

To find the forward messages, we can exploit the fact that our transition functions are Gaussian and can therefore be reversed. This means that given a transition $\phi(y|x)$, we can find a reversed transition $\phi^T(x|y)$ that evaluates to the same value as $\phi(y|x)$ for all $x, y$

**Definition 6** (Reversed transition). *Let $\phi(y|x)$ be a transition function. Then the reversed transition is defined as*

$$\phi^T(x|y) = \phi(y|x) \tag{34}$$

*so that $\phi^T(x|y) = \phi(y|x)$ for all $x, y$ and $\int \phi^T(x|y)dx = \int \phi(y|x)dx = 1$.*

Using this reverse operation, we can simply reverse the transition distributions and then find the forward messages by using the same recurrence relation as for the backward messages:

$$\int \underbrace{\phi_{k+1|k}(x_{k+1}|x_k)}_{\text{reverse}} \underbrace{\phi(x_k|\theta_k)\phi(x_k|\alpha_k)}_{\text{multiply} \rightarrow \phi(x_k|\theta_k+\alpha_k)} dx_k \tag{35}$$

$$= \int \underbrace{\phi^T(x_k|x_{k+1})\phi(x_k|\theta_k + \alpha_k)}_{\text{update} \rightarrow \hat{\phi}^T(x_k|x_{k+1})\phi(x_{k+1}|\alpha_{k+1})} dx_k \tag{36}$$

$$= \underbrace{\int \hat{\phi}^T(x_k|x_{k+1})dx_k}_{\text{transition integrates to 1}} \phi(x_{k+1}|\alpha_{k+1}) \tag{37}$$

$$= \phi(x_{k+1}|\alpha_{k+1}) \tag{38}$$

These message passing updates can be computed in $O(N)$ time using the the multiply, update and reverse operations. However, there is a more efficient way to compute the forward messages using the parallel scan algorithm [Särkkä and García-Fernández, 2020] that reduces the complexity to $O(\log N)$ on parallel compute. We will describe this algorithm in Appendix D.2.

## D.2 Parallel message passing

In this section we will use slightly different notation to describe the parallel message passing algorithm. We will avoid writing out the parameters of our potential functions and call them by their parameter name. For example, instead of writing $\phi(x_k|\theta_k)$, we will write $\phi_k(x_k)$ and instead of writing $\phi(x_k|\beta_k)$, we will write $\beta(x_k)$.

The building block of the parallel message passing algorithm Särkkä and García-Fernández [2020] is an unnormalized potential function over two variables, which we denote by $\Psi(y, x)$. We assume that $\Psi(y, x)$ can be decomposed into a (normalized) transition distribution and an unnormalized potential function:

$$\Psi(y, x) = \Psi(y|x)\Psi(x) \tag{39}$$

Whenever we write $\Psi(y|x)$, we are referring to a valid conditional probability distribution ($\int \Psi(y|x)dy = 1$). Since $\Psi(y, x)$ is jointly Gaussian over $x$ and $y$, we are able to integrate out variables in $x$ and $y$ and can also combine neighboring potentials into a new Gaussian potential. These properties allow us to construct a chain operation over potentials that combines neighboring potentials and then integrates out the common variable. We denote this chain operation by $\otimes$:

$$\Psi(y, x) := \int \Psi(y, z)\Psi(z, x)dz =: \Psi(y, z) \otimes \Psi(z, x) \tag{40}$$

An important property of the chain operation is that it is associative due to the fact that we can swap the order or integration (we will prove this in Appendix D.3).

A useful perspective of this chain operation is that it amounts to performing variable elimination on the graph defined by the potentials, i.e. performs some sort of message passing [Koller, 2009]. With this in mind, we can perform message passing by constructing the appropriate joint potentials:

**Proposition 9** (Parallel messages). *Let $\phi_{k+1|k}$ and $\phi_k$ be the potential functions for the CRF in Definition 1 and $\alpha$ and $\beta$ be the messages defined in Eqs. (26) and (33). Then*

$$\alpha_k(x_k) = \int \Psi_{1:k}^{fwd}(x_k, x_1)dx_1 \quad and \quad \beta_k(x_k) = \int \Psi_{k:N}^{bwd}(x_N|x_k)dx_N \tag{41}$$

*where*

$$\Psi_{1:k}^{fwd}(x_k, x_1) = \bigotimes_{i=1}^{k-1} \phi_{i+1|i}(x_{i+1}|x_i)\phi_i(x_i) \tag{42}$$

$$and \quad \Psi_{k:N}^{bwd}(x_N|x_k) = \bigotimes_{i=N-1}^{k} \phi_{i+1|i}(x_{i+1}|x_i)\phi_{i+1}(x_{i+1}) \tag{43}$$

See appendix Appendix D.3 for a proof and **??** for pseudocode. Since $\otimes$ is associative, we can evaluate Eq. (42) in $O(\log N)$ time using the parallel scan algorithm [Särkkä and García-Fernández, 2020]. The rough idea is that on parallel compute, one can, in parallel, chain together consecutive pairs of potentials and then recurse on these new chained potentials in order to eventually chain the entire sequence. We provide pseudocode for this a special case of this algorithm in Appendix H.3. $\Psi_{1:k}^{fwd}(x_k, x_1)$ and $\Psi_{k:N}^{bwd}(x_N|x_k)$ can be thought of as the result of marginalization over the variables between $x_1$ and $x_k$ and $x_k$ and $x_N$, respectively.

## D.3 Chain operation

Recall that the chain operation is defined in Eq. (40) as

$$\Psi(y, x) := \int \Psi(y, z)\Psi(z, x)dz =: \Psi(y, z) \otimes \Psi(z, x) \tag{44}$$

To see that it is associative, we need to check that $\Psi(y, z) \otimes (\Psi(z, x) \otimes \Psi(x, w)) = (\Psi(y, z) \otimes \Psi(z, x)) \otimes \Psi(x, w)$

$$\Psi(y, z) \otimes (\Psi(z, x) \otimes \Psi(x, w)) = \int \Psi(y, z) \left( \int \Psi(z, x)\Psi(x, w)dx \right) dz \tag{45}$$

$$= \int \int \Psi(y, z)\Psi(z, x)\Psi(x, w)dxdz \tag{46}$$

$$= \int \left( \int \Psi(y, z)\Psi(z, x)dz \right) \Psi(x, w)dx \tag{47}$$

$$= (\Psi(y, z) \otimes \Psi(z, x)) \otimes \Psi(x, w) \tag{48}$$

**Proposition 10** (Parallel messages). *Let $\phi_{k+1|k}$ and $\phi_k$ be the potential functions for the CRF in Definition 1 and $\alpha$ and $\beta$ be the messages defined in Eqs. (26) and (33). Then*

$$\alpha_k(x_k) = \int \Psi_{1:k}^{fwd}(x_k, x_1)dx_1 \quad and \quad \beta_k(x_k) = \int \Psi_{k:N}^{bwd}(x_N|x_k)dx_N \tag{49}$$

*where*

$$\Psi_{1:k}^{fwd}(x_k, x_1) = \bigotimes_{i=1}^{k-1} \phi_{i+1|i}(x_{i+1}|x_i)\phi_i(x_i) \tag{50}$$

$$and \quad \Psi_{k:N}^{bwd}(x_N|x_k) = \bigotimes_{i=N-1}^{k} \phi_{i+1|i}(x_{i+1}|x_i)\phi_{i+1}(x_{i+1}) \tag{51}$$

*Proof.* First for notational clarity, define

$$\Psi_{i+1,i}^{\text{bwd}}(x_{i+1}|x_i) = \phi_{i+1|i}(x_{i+1}|x_i)\phi_{i+1}(x_{i+1}) \quad \text{and} \quad \Psi_{i+1,i}^{\text{fwd}}(x_{i+1}, x_i) = \phi_{i+1|i}(x_{i+1}|x_i)\phi_i(x_i) \tag{52}$$

We can compute the cumulative potentials as follows:

$$\Psi_{k:N}^{\text{bwd}}(x_N|x_k) = \bigotimes_{i=N-1}^{k} \Psi_{i+1,i}^{\text{bwd}}(x_{i+1}|x_i) \tag{53}$$

$$= \Psi_{N:N-1}^{\text{bwd}}(x_N|x_{N-1}) \otimes \Psi_{N-1:N-2}^{\text{bwd}}(x_{N-1}|x_{N-2}) \otimes \cdots \otimes \Psi_{k+1:k}^{\text{bwd}}(x_{k+1}|x_k) \tag{54}$$

$$= \int \Psi_{N:N-1}^{\text{bwd}}(x_N|x_{N-1}) \int \Psi_{N-1:N-2}^{\text{bwd}}(x_{N-1}|x_{N-2})dx_{N-1} \int \Psi_{N-2:N-3}^{\text{bwd}}(x_{N-2}|x_{N-3})dx_{N-2} \cdots dx_{k+1} \tag{55}$$

$$= \int \cdots \int \prod_{i=k}^{N-1} \Psi_{i:i+1}^{\text{bwd}}(x_{i+1}|x_i)dx_{N-1} \cdots dx_{k+1} \tag{56}$$

And similarly for the forward potentials:

$$\Psi_{1:k}^{\text{fwd}}(x_k, x_1) = \bigotimes_{i=1}^{k-1} \Psi_{i+1,i}^{\text{fwd}}(x_{i+1}, x_i) \tag{57}$$

$$= \int \cdots \int \prod_{i=1}^{k-1} \Psi_{i+1,i}^{\text{fwd}}(x_{i+1}, x_i)dx_2 \cdots dx_{k-1} \tag{58}$$

Next, we can rewrite the joint distribution of the CRF in a similar form:

$$p(x_{1:N}) = \prod_{k=1}^{N-1} \phi_{k+1|k}(x_{k+1}|x_k) \prod_{k=1}^{N} \phi_k(x_k) \tag{59}$$

$$= \phi_k(x_k) \prod_{i=k}^{N-1} \Psi_{i+1,i}^{\text{bwd}}(x_{i+1}|x_i) \prod_{i=1}^{k-1} \Psi_{i+1,i}^{\text{fwd}}(x_{i+1}, x_i), \quad \forall k \in \{1, \ldots, N\} \tag{60}$$

Then, integrating over the variables $dx_1, \ldots, \hat{dx}_k, \ldots, dx_N$, where $\hat{dx}_k$ denotes that we are not integrating over $x_k$, completes the proof:

$$p(x_k) = \int \cdots \int p(x_{1:N})dx_1 \ldots \hat{dx}_k \ldots dx_N \tag{61}$$

$$\propto \int \cdots \int \prod_{k=1}^{N-1} \phi_{k+1|k}(x_{k+1}|x_k) \prod_{k=1}^{N} \phi_k(x_k)dx_1 \ldots \hat{dx}_k \ldots dx_N \tag{62}$$

$$= \phi_k(x_k) \int \cdots \int \prod_{i=k}^{N-1} \Psi_{i+1,i}^{\text{bwd}}(x_{i+1}|x_i) \prod_{i=1}^{k} \Psi_{i+1,i}^{\text{fwd}}(x_{i+1}, x_i)dx_1 \ldots \hat{dx}_k \ldots dx_N \tag{63}$$

$$= \phi_k(x_k) \underbrace{\int \Psi_{k:N}^{\text{bwd}}(x_N|x_k)dx_N}_{\beta_k(x_k)} \underbrace{\int \Psi_{1:k}^{\text{fwd}}(x_k, x_1)dx_1}_{\alpha_k(x_k)} \tag{64}$$

We can recognize the terms in the last equation as the forward and backward messages, which completes the proof. □

It will be convenient later to define an operator that actually transforms the parameters of the backward messages.

**Definition 7** (Message passing update operator). *Let $\phi_{k+1|k}(x_{k+1}, x_k)$ be a Gaussian transition function and let $\phi(x_{k+1}|\eta_{k+1})$ be a Gaussian node potential with natural parameters $\eta_{k+1}$. Next consider the message passing update:*

$$\phi(x_k|\eta_k) = \int \phi_{k+1|k}(x_{k+1}|x_k)\phi(x_{k+1}|\eta_{k+1})dx_{k+1} \tag{65}$$

*The message passing update operator is denoted by $\Phi_{k,k+1}(\eta_{k+1})$ and is defined to satisfy:*

$$\eta_k = \Phi_{k,k+1}(\eta_{k+1}) \tag{66}$$

*In particular, the update rule for the backward messages is given by:*

$$\beta_k = \Phi_{k,k+1}(\beta_{k+1} + \theta_{k+1}) \tag{67}$$

**Corollary 2** (Mixed parameterization update rule). *Let $\phi_{k+1|k}(x_{k+1}|x_k) := N(x_{k+1}|Ax_k+u, \Sigma)$ be a Gaussian transition function and let $\phi(x_{k+1}|\eta_{k+1}) := N(x_{k+1}|\mu_{k+1}, J_{k+1}^{-1})$ be a Gaussian node potential where $J_{k+1}$ is the precision matrix. If $\eta_k$ and $\eta_{k+1}$ represent the mean and precision matrix of a Gaussian distribution, then the update and marginalize operator is denoted by $\Phi_{k,k+1}(\eta_{k+1})$ and is given by:*

$$\Phi_{k,k+1}\left(\mu_{k+1}, J_{k+1}\right) = \left(A^{-1}(\mu_{k+1} - u), \Phi_{k,k+1}^{(J)}(J_{k+1})\right) \tag{68}$$

*where $\Phi_{k,k+1}^{(J)}(J_{k+1})$ is a nonlinear function of $J_{k+1}$.*

*Proof.* The result follows from Appendix H.3. □

### D.4 Probabilistic queries

The forward and backward messages can be used to compute the majority of the probabilistic queries of interest on a CRF. Recall our definition of a CRF:

$$p(x_{1:N}|\theta) \propto \prod_{k=1}^{N-1} \phi_{k+1|k}(x_{k+1}|x_k) \prod_{k=1}^{N} \phi(x_k|\theta_k) \tag{69}$$

Next we will describe two probabilistic queries of interest: the marginal distribution and the transition distribution.

**Proposition 11** (Marginal distribution).
$$p(x_k|\theta) = \phi(x_k|\theta_k + \alpha_k + \beta_k) \tag{70}$$

*Proof.* The derivation is given in Eq. (61). For completeness, we will change notation:

$$p(x_k) = \phi_k(x_k)\beta_k(x_k)\alpha_k(x_k) \text{ (notation in previous section)} \tag{71}$$
$$:= \phi(x_k|\theta_k)\phi(x_k|\alpha_k)\phi(x_k|\beta_k) \text{ (notation in this section and in main text)} \tag{72}$$
$$= \phi(x_k|\theta_k + \alpha_k + \beta_k) \tag{73}$$

□

**Proposition 12** (Transition distribution).
$$p(x_{k+1}|x_k, \theta) \propto \phi_{k+1|k}(x_{k+1}|x_k)\phi(x_{k+1}|\theta_{k+1} + \beta_{k+1}) \tag{74}$$

*Proof.* We can start by computing the joint distribution $p(x_{k+1}, x_k|\theta)$. By using variable elimination, we can show that

$$p(x_{k+1}, x_k|\theta) = \phi(x_k|\alpha_k)\phi_{k+1|k}(x_{k+1}|x_k)\phi(x_{k+1}|\theta_{k+1})\phi(x_{k+1}|\beta_{k+1}) \tag{75}$$

Dividing by the marginal distribution $p(x_k|\theta)$ and using the definition of the transition distribution, we get

$$p(x_{k+1}|x_k, \theta) = \phi_{k+1|k}(x_{k+1}|x_k)\frac{\phi(x_{k+1}|\beta_{k+1} + \theta_{k+1})}{\phi(x_k|\beta_k + \theta_k)} \tag{76}$$

which, after absorbing the denominator into the normalization constant, is equivalent to the desired result. □

**Corollary 3** (Autoregressive factorization). *The autoregressive factorization of $p(x_{1:N}|\theta)$ takes the following form:*

$$p(x_{1:N}|\theta) \propto \phi(x_1|\theta_1 + \beta_1) \prod_{k=1}^{N-1} \phi_{k+1|k}(x_{k+1}|x_k)\phi(x_{k+1}|\theta_{k+1} + \beta_{k+1}) \tag{77}$$

*Proof.* This follows directly from applying Proposition 11 and Proposition 12 to $p(x_{1:N}|\theta) = p(x_1|\theta)\prod_{k=1}^{N-1} p(x_{k+1}|x_k, \theta)$. $\qquad\square$

# E  Conditioned SDEs

In this section we derive the form of conditioned linear SDEs as well as the corresponding probability flow ODEs.

## E.1  Conditioned linear SDE

**Proposition 13** (Conditioned Linear SDE). *Let $\phi_{t+s|t}(x_{t+s}|x_t)$ be the transition distribution of the linear SDE $dx_t = F_t x_t dt + L_t dW_t$ and let $\{\phi(x_{t_k}|\theta_{t_k})\}_{t_k \in \mathcal{R}}$ be potential functions at times in the set $\mathcal{R}$. Then the piecewise-linear SDE,*

$$dx_t = (F_t x_t + L_t L_t^T \nabla \log \phi(x_t|\beta_t))dt + L_t dW_t, \quad x_{t_1} \sim \phi(x_{t_1}|\beta_1 + \theta_1) \tag{78}$$

*where $t \in (t_k, t_{k+1})$ and $t_k, t_{k+1} \in \mathcal{R}$, has a joint distribution over any superset of times $t_{1:N} = \mathcal{T} \supseteq \mathcal{R}$ that is given by a CRF:*

$$p(x_{t_{1:N}}|\theta) \propto \prod_{t_k \in \mathcal{T}} \phi_{t_{k+1}|t_k}(x_{t_{k+1}}|x_{t_k}) \prod_{t_k \in \mathcal{R}} \phi(x_{t_k}|\theta_{t_k}) \tag{79}$$

*where $\beta_t$ is the extension of the backward message defined in **??** to time $t$:*

$$\phi(x_t|\beta_t) = \int \phi_{t_{k+1}|t}(x_{t_{k+1}}|x_t)\phi(x_{t_{k+1}}|\theta_{t_{k+1}} + \beta_{t_{k+1}})dx_{t_{k+1}} \tag{80}$$

*Proof.* We will first construct the transition distribution of the conditioned SDE and then use Doob's h-transform to identify the form of the SDE. Recall that Doob's h-transform ([Särkkä and Solin, 2019] section 7.5) is used to find the SDE associated with a transition distribution of the form $p(x_{t+s}|x_t) = \phi_{t+s|t}(x_{t+s}|x_t)\frac{h_{t+s}(x_{t+s})}{h_t(x_t)}$ where $\phi_{t+s|t}(x_{t+s}|x_t)$ is the transition distribution of a base SDE with the form $dx_t = u_t dt + L_t dW_t$ and $h_t$ is a function that satisfies $h_t(x_t) = \int_t^{t+s} \phi_{t+s|t}(x_{t+s}|x_t)h_{t+s}(x_{t+s})dx_{t+s}$. Then the SDE whose transition distribution is $p(x_{t+s}|x_t)$ is given by

$$dx_t = (u_t + L_t L_t^T \nabla \log h_t(x_t))dt + L_t dW_t \tag{81}$$

We will show that the backward messages of the CRF are of the form $h_t(x_t)$ and then use Doob's h-transform to identify the form of the conditioned SDE.

Suppose $t \in (t_k, t_{k+1})$ and $s > 0$ is small enough so that $t + s \in (t_k, t_{k+1})$. Then we can construct the joint distribution over $(t_{t+s}, t_{k+1}, \ldots, t_N)$ given $x_t$ as

$$p(x_{t+s}|x_t) = \int \cdots \int p(x_{t_{k+1:N}}, x_{t+s}|x_t)dx_{t_{k+1}} \cdots dx_{t_N} \tag{82}$$

$$\propto \int \cdots \int \phi(x_{t_{k+1}}|\theta_{t_{k+1}}) \underbrace{\left( \prod_{i=k+1}^{N-1} \phi_{t_{i+1}|t_i}(x_{t_{i+1}}|x_{t_i})\phi(x_{t_{i+1}}|\theta_{t_{i+1}}) \right) \phi_{t_{k+1}|t+s}(x_{t_{k+1}}|x_{t+s})dx_{t_{k+1}}}_{\text{integrate to get parallel bwd message (Proposition 9)}} \cdots dx_{t_N}\phi_{t+s|t}(x_{t+s}|x_t)$$
$$\tag{83}$$

$$= \int \int \phi(x_{t_{k+1}}|\theta_{t_{k+1}})\Psi_{k+1:N}^{\text{bwd}}(x_{t_N}|x_{t_{k+1}})\phi_{t_{k+1}|t+s}(x_{t_{k+1}}|x_{t+s})dx_{t_N}dx_{t_{k+1}}\phi_{t+s|t}(x_{t+s}|x_t) \tag{84}$$

$$= \underbrace{\int \phi(x_{t_{k+1}}|\theta_{t_{k+1}})\phi(x_{t_{k+1}}|\beta_{t_{k+1}})\phi_{t_{k+1}|t+s}(x_{t_{k+1}}|x_{t+s})dx_{t_{k+1}}}_{=:\phi(x_{t+s}|\beta_{t+s})} \phi_{t+s|t}(x_{t+s}|x_t) \tag{85}$$

$$= \phi(x_{t+s}|\beta_{t+s})\phi_{t+s|t}(x_{t+s}|x_t) \tag{86}$$

We can find the normalizing constant by integrating over $x_{t+s}$:

$$\int \phi(x_{t+s}|\beta_{t+s})\phi_{t+s|t}(x_{t+s}|x_t)dx_{t+s} \tag{87}$$

$$= \int\int \phi(x_{t_{k+1}}|\theta_{t_{k+1}})\phi(x_{t_{k+1}}|\beta_{t_{k+1}})\phi_{t_{k+1}|t+s}(x_{t_{k+1}}|x_{t+s})dx_{t_{k+1}}\phi_{t+s|t}(x_{t+s}|x_t)dx_{t+s} \tag{88}$$

$$= \int \phi(x_{t_{k+1}}|\theta_{t_{k+1}})\phi(x_{t_{k+1}}|\beta_{t_{k+1}})\underbrace{\int \phi_{t_{k+1}|t+s}(x_{t_{k+1}}|x_{t+s})\phi_{t+s|t}(x_{t+s}|x_t)dx_{t+s}}_{\phi_{t_{k+1}|t}(x_{t_{k+1}}|x_t)}dx_{t_{k+1}} \tag{89}$$

$$= \int \phi(x_{t_{k+1}}|\theta_{t_{k+1}})\phi(x_{t_{k+1}}|\beta_{t_{k+1}})\phi_{t_{k+1}|t}(x_{t_{k+1}}|x_t)dx_{t_{k+1}} \tag{90}$$

$$= \phi(x_t|\beta_t) \tag{91}$$

Therefore, the transition distribution is

$$p(x_{t+s}|x_t) = \phi_{t+s|t}(x_{t+s}|x_t)\frac{\phi(x_{t+s}|\beta_{t+s})}{\phi(x_t|\beta_t)} \tag{92}$$

Note that Eq. (87) also verifies that $\phi(x_t|\beta_t)$ satisfies the normalization condition for $h_t(x_t)$ in Doob's h-transform. Directly applying Doob's h-transform to the transition distribution in Eq. (82) identifies the form of the conditioned SDE:

$$dx_t = (F_t x_t + L_t L_t^T \nabla \log \phi(x_t|\beta_t))dt + L_t dW_t \tag{93}$$

This piecewise-linear SDE has the correct conditional distribution, $p(x_t|x_{t_{k_1}})$, but requires an initial distribution. One can verify that the initial distribution $p(x_{t_1}) \propto \phi(x_{t_1}|\theta_{t_1} + \beta_{t_1})$ is the first marginal distribution of the CRF in Definition 1. $\qquad\square$

### E.2 Probabilistic queries for conditioned linear SDEs

**Lemma 2** (Marignal distribution of conditioned SDE). *Suppose $t \in (t_k, t_{k+1})$ is a time in between the inducing points $t_k$ and $t_{k+1}$ of the conditioned linear SDE in Proposition 4. Then the marginal distribution of the SDE at time $t$ is given by*

$$p(x_t) = \phi(x_t|\alpha_t + \beta_t) \tag{94}$$

*where $\alpha_t$ and $\beta_t$ are extensions of the forward and backward messages defined in Eq. (33) and Eq. (26) to time $t$:*

$$\phi(x_t|\alpha_t) = \int \phi_{t|t_{k-1}}(x_t|x_{t_{k-1}})\phi(x_{t_{k-1}}|\theta_{t_{k-1}} + \alpha_{t_{k-1}})dx_{t_{k-1}} \tag{95}$$

*and*

$$\phi(x_t|\beta_t) = \int \phi_{t|t_{k+1}}(x_t|x_{t_{k+1}})\phi(x_{t_{k+1}}|\theta_{t_{k+1}} + \beta_{t_{k+1}})dx_{t_{k+1}} \tag{96}$$

*Proof.* We can simply incorporate $t$ into the set discretization times, $t_{1:N}$, used in Proposition 4 to get the desired result. Suppose $t \in (t_i, t_{i+1})$ for some $i$. Then we can write the joint distribution as

$$p(x_t, x_{t_{1:N}}|\theta) \propto \phi_{t_{i+1}|t_i}(x_{t_{i+1}}|x_{t_i})\phi_{t|t_i}(x_t|x_{t_i})\prod_{t_k \in \mathcal{T}}\phi_{t_{k+1}|t_k}(x_{t_{k+1}}|x_{t_k})\prod_{t_k \in \mathcal{R}}\phi(x_{t_k}|\theta_{t_k}) \tag{97}$$

Then we can run variable elimination on the ends of the chain until we are left with the marginal distribution of $x_t$:

$$p(x_t) = \int p(x_t, x_{t_{1:N}}|\theta)dx_{t_{1:N}} \tag{98}$$

$$= \int\int \phi(x_{t_i}|\alpha_{t_i} + \theta_{t_i})\phi_{t|t_i}(x_t|x_{t_i})\phi_{t_{i+1}|t}(x_{t_{i+1}}|x_t)\phi(x_{t_{i+1}}|\beta_{t_{i+1}} + \theta_{t_{i+1}})dx_{t_{i+1}}dx_{t_i} \tag{99}$$

$$= \underbrace{\int \phi(x_{t_i}|\alpha_{t_i} + \theta_{t_i})\phi_{t|t_i}(x_t|x_{t_i})dx_{t_i}}_{\phi(x_t|\alpha_t)} \underbrace{\int \phi_{t_{i+1}|t}(x_{t_{i+1}}|x_t)\phi(x_{t_{i+1}}|\beta_{t_{i+1}} + \theta_{t_{i+1}})dx_{t_{i+1}}}_{\phi(x_t|\beta_t)}$$

(100)

$$= \phi(x_t|\alpha_t + \beta_t) \tag{101}$$

$\square$

**Lemma 3** (Transition distribution of conditioned linear SDE). *Suppose $t \in (t_k, t_{k+1})$ is a time in between the inducing points $t_k$ and $t_{k+1}$ of the conditioned linear SDE in Proposition 4, and suppose that $s > 0$ is small enough so that $t + s \in (t_k, t_{k+1})$. Then the transition distribution of the SDE at time $t$ is given by*

$$\phi_{t+s|t}(x_{t+s}|x_t) \propto \phi_{t+s|t}(x_{t+s}|x_t)\phi(x_{t+s}|\beta_{t+s}) \tag{102}$$

*Proof.* The proof is embedded in the derivation of the conditioned linear SDE at Eq. (92). $\square$

**Corollary 4** (Autoregressive factorization). *The autoregressive factorization of $p(x_{t_{1:N}}|\theta)$ is given by*

$$p(x_{t_{1:N}}|\theta) = p(x_{t_1}|\theta) \prod_{t_k \in \mathcal{T}} \phi_{t_k|t_{k-1}}(x_{t_k}|x_{t_{k-1}})\phi(x_{t_k}|\beta_{t_k}) \tag{103}$$

$$where \ \beta_{t_k} = \begin{cases} \Phi_{t_k,t_{k+1}}(\beta_{t_{k+1}} + \theta_{t_{k+1}}) & if \ t_k \in \mathcal{R} \\ \Phi_{t_k,t_{k+1}}(\beta_{t_{k+1}}) & otherwise \end{cases} \tag{104}$$

*where $\Phi_{t_k,t_{k+1}}$ is the message passing update operator defined in Definition 7.*

*Proof.* Recall that

$$p(x_{t_{1:N}}|\theta) \propto \prod_{t_k \in \mathcal{T}} \phi_{t_{k+1}|t_k}(x_{t_{k+1}}|x_{t_k}) \prod_{t_k \in \mathcal{R}} \phi(x_{t_k}|\theta_{t_k}) \tag{105}$$

Suppose that for each $t_k \notin \mathcal{R}$, we introduce a new potential function whose natural parameters are 0, which we will denote by $\phi(x_{t_k}|\emptyset_{t_k})$. These new potentials have no effect on the joint distribution, but allow us to rewrite the joint distribution in the same form as in Corollary 3, which yields the result. $\square$

### E.3 Probability flow ODE for conditioned linear SDEs

**Corollary 5** (Probability flow ODE). *The probability flow ODE of the SDE in Proposition 4 is given by*

$$\frac{dx_t}{dt} = F_t x_t + \frac{1}{2}L_t L_t^T \left(\nabla \log \phi(x_t|\beta_t) - \nabla \log \phi(x_t|\alpha_t)\right) \tag{106}$$

*$\beta_t$ is the same as in Proposition 4 and $\alpha_t$ is the extension of the forward message defined in Eq. (33) to time $t$:*

$$\phi(x_t|\alpha_t) = \int \phi_{t|t_k}(x_t|x_{t_k})\phi(x_{t_k}|\theta_{t_k} + \alpha_{t_k})dx_{t_k} \tag{107}$$

*Proof.* Let $dx_t = u_t dt + L_t dW_t$ be an SDE. Then the probability flow ODE is defined Song et al. [2021] as

$$\frac{dx_t}{dt} = u_t - \frac{1}{2}L_t L_t^T \nabla \log p_t(x_t) \tag{108}$$

where $p_t(x_t)$ is defined as the marginal distribution of the SDE, which is given by Lemma 2. We can apply this directly to our SDE in Proposition 4 to get the result:

$$\frac{dx_t}{dt} = (F_t x_t + L_t L_t^T \nabla \log \phi(x_t|\beta_t)) - \frac{1}{2}L_t L_t^T \nabla \log p_t(x_t) \tag{109}$$

$$= (F_t x_t + L_t L_t^T \nabla \log \phi(x_t|\beta_t)) - \frac{1}{2}L_t L_t^T \left(\nabla \log \phi(x_t|\alpha_t) + \nabla \log \phi(x_t|\beta_t)\right) \tag{110}$$

$$= F_t x_t + \frac{1}{2}L_t L_t^T \left(\nabla \log \phi(x_t|\beta_t) - \nabla \log \phi(x_t|\alpha_t)\right) \tag{111}$$

$\square$

 # F   CMFVI proofs

 ## F.1   Constrained mean field VI

Let $\theta \sim p(\theta)$ be an unknown prior distribution on the parameters of the conditional exponential family distribution, $p(x|z, \theta) \propto \exp\{\langle t_z(x), \theta \rangle - A(z, \theta)\}$, where $t_z(x)$ is the sufficient statistic of the exponential family distribution and $A(z, \theta)$ is the log partition function. In our setting, we interpret $x$ and $z$ as unobserved and observed variables and $\theta$ as a a a parameter that they both depend on. We are interested in performing inference in the predictive distribution $p(x|z)$, where we must integrate out $\theta$. This distribution can be written as:

$$p(x|z) = \int p(x|z, \theta)p(\theta|z)d\theta \tag{112}$$

$$= \mathbb{E}_{p(\theta|z)} \left[ \exp\{\langle t_z(x), \theta \rangle - A(z, \theta)\} \right] \tag{113}$$

where $t_z(x)$ is the sufficient statistic of the conditional exponential family distribution. Since this distribution is intractable, we use a variational approximation to approximate it. Our variational approximation is called the constrained mean field VI approximation and is given by:

$$q^*(x|z) = \underset{q(x|z)}{\operatorname{argmin}} \ \mathrm{KL}\left[ q(x|z)p(\theta|z) \| p(x, \theta|z) \right] \tag{114}$$

In this appendix section we will derive facts about $q^*(x|z)$.

**Lemma 4** (Alternate constrained mean field VI objectives). *The constrained mean field VI objective,*

$$\mathrm{KL}\left[ q(x|z)p(\theta|z) \| p(x, \theta|z) \right] \tag{115}$$

*is equal to the following expressions:*

*1.*

$$\mathbb{E}_{q(x|z)\,p(\theta|z)} \left[ \log \frac{p(\theta|z)}{p(\theta|x, z)} \right] + \mathrm{KL}\left[ q(x|z) \| p(x|z) \right] \tag{116}$$

*2.*

$$\mathbb{E}_{q(x|z)\,p(\theta|z)} \left[ \log \frac{p(x|z)}{p(x|z, \theta)} \right] + \mathrm{KL}\left[ q(x|z) \| p(x|z) \right] \tag{117}$$

*3.*

$$\mathbb{E}_{q(x|z)} \left[ \log q(x|z) - \mathbb{E}_{p(\theta|z)} \left[ \log p(x|z, \theta) \right] \right] \tag{118}$$

*Proof.* The proof is a straightforward rearrangement of terms:

$$\mathrm{KL}\left[ q(x|z)p(\theta|z) \| p(x, \theta|z) \right] = \int\int q(x|z)p(\theta|z) \log \frac{q(x|z)p(\theta|z)}{p(x, \theta|z)} dx dy \tag{119}$$

$$= \int\int q(x|z)p(\theta|z) \log \frac{p(\theta|z)}{p(\theta|x, z)} \frac{q(x|z)}{p(x|z)} dx dy \quad \text{(equals 1)} \tag{120}$$

$$= \int\int q(x|z)p(\theta|z) \log \frac{\cancel{p(x|z)}}{p(x|z, \theta)} \frac{q(x|z)}{\cancel{p(x|z)}} dx dy \quad \text{(equals 2)} \tag{121}$$

$$= \int\int q(x|z)p(\theta|z) \log \frac{q(x|z)}{p(x|z, \theta)} dx dy \tag{122}$$

$$= \mathbb{E}_{q(x|z)} \left[ \log q(x|z) - \mathbb{E}_{p(\theta|z)} \left[ \log p(x|z, \theta) \right] \right] \tag{123}$$

$\square$

**Theorem 2** (Constrained mean field VI solution). *Let $p(x|z, \theta) \propto \exp\{\langle t_z(x), \theta \rangle - A(z, \theta)\}$ be an exponential family distribution and that $\theta \sim p(\theta|z)$. The constrained mean field VI approximation of $p(x|z)$, denoted by $q^*(x|z)$, is defined as follows:*

$$q^*(x|z) = \underset{q(x|z)}{\operatorname{argmin}} \mathrm{KL}\left[ q(x|z)p(\theta|z) \| p(x, \theta|z) \right] \tag{124}$$

$$= p(x|z, \theta^*(z)), \quad where \ \theta^*(z) = \mathbb{E}_{p(\theta|z)} \left[ \theta \right] \tag{125}$$

*Proof.* The proof can follow quickly from the standard mean field VI solutions Beal [2003], but for completeness we will derive it from scratch. Starting from the result of Lemma 4, we have that

$$q^*(x|z) = \underset{q(x|z)}{\operatorname{argmin}} \ \mathbb{E}_{q(x|z)} \left[ \log q(x|z) - \mathbb{E}_{p(\theta|z)} \left[ \log p(x|z,\theta) \right] \right] \tag{126}$$

We can introduce a Lagrange multiplier to enforce the constraint that the distribution is normalized. Let $q_\epsilon(x|z) = q(x|z) + \epsilon\eta(x|z)$ where $\eta$ is the variation function and $\epsilon$ is a scalar. Then we can take a variation by differentiating with respect to $\epsilon$:

$$\frac{\partial}{\partial \epsilon} \left( \mathbb{E}_{q_\epsilon(x|z)} \left[ \log q_\epsilon(x|z) - \mathbb{E}_{p(\theta|z)} \left[ \log p(x|z,\theta) \right] \right] + \lambda \left( \int q_\epsilon(x|z)dx - 1 \right) \right) = 0 \tag{127}$$

$$\implies \frac{\partial}{\partial \epsilon} \int q_\epsilon(x|z) \log q_\epsilon(x|z)dx + \int \eta(x|z) \left( \mathbb{E}_{p(\theta|z)} \left[ \log p(x|z,\theta) \right] + \lambda \right) dx = 0 \tag{128}$$

The negative entropy term simplies as follows:

$$\frac{\partial}{\partial \epsilon} \int q_\epsilon(x|z) \log q_\epsilon(x|z)dx = \int \frac{\partial}{\partial \epsilon} q_\epsilon(x|z) \log q_\epsilon(x|z)dx + \int q_\epsilon(x|z)\frac{\partial}{\partial \epsilon} \log q_\epsilon(x|z)dx \tag{129}$$

$$= \int \frac{\partial q_\epsilon(x|z)}{\partial \epsilon} \log q_\epsilon(x|z)dx + \int q_\epsilon(x|z)\frac{\partial \log q_\epsilon(x|z)}{\partial \epsilon}dx \tag{130}$$

$$= \int \eta(x|z) \log q_\epsilon(x|z)dx - \int q_\epsilon(x|z)\frac{1}{q_\epsilon(x|z)}\frac{\partial q_\epsilon(x|z)}{\partial \epsilon}dx \tag{131}$$

$$= \int \eta(x|z) \left( \log q_\epsilon(x|z) - 1 \right) dx \tag{132}$$

Plugging this back into the original equation and setting it equal to zero implies that the integrand must be zero:

$$\mathbb{E}_{p(\theta|z)} \left[ \log p(x|z,\theta) \right] + \lambda + \log q_\epsilon(x|z) - 1 = 0 \tag{133}$$

Solving for $\log q_\epsilon(x|z)$ (and setting $\epsilon = 0$) yields:

$$\log q(x|z) = \mathbb{E}_{p(\theta|z)} \left[ \log p(x|z,\theta) \right] + \lambda - 1 \tag{134}$$

The lagrange multiplier $\lambda$ ensures that the distribution is normalized, and so we have that

$$q^*(x|z) = \exp \left\{ \mathbb{E}_{p(\theta|z)} \left[ \log p(x|z,\theta) \right] + \lambda - 1 \right\} \tag{135}$$

$$\propto \exp \left\{ \mathbb{E}_{p(\theta|z)} \left[ \log p(x|z,\theta) \right] \right\} \tag{136}$$

$$\propto \exp \left\{ \langle t_z(x), \mathbb{E}_{p(\theta|z)} [\theta] \rangle \right\} \tag{137}$$

And so we can recognize that $q^*(x|z)$ is in the same exponential family as $p(x|z,\theta)$ but with natural parameter $\mathbb{E}_{p(\theta|z)} [\theta]$. This completes the proof. $\qquad \square$

Next, we emphasize another form of the CMFVI solution that is convenient when deriving CMFVI solutions of other models.

**Lemma 5** (Mean field form of CMFVI solution). *The CMFVI approximation of $p(x|z)$ has the following form:*

$$q^*(x|z) \propto \exp \left\{ \mathbb{E}_{p(\theta|z)} \left[ \log p(x|z,\theta) \right] \right\} \tag{138}$$

*Proof.* See Eq. (136) $\qquad \square$

**Corollary 6** (Value of CMFVI objective at optimum). *The value of the CMFVI objective at the optimum is given by:*

$$\mathrm{KL} \left[ q^*(x|z)p(\theta|z) \| p(x,\theta|z) \right] = \mathbb{E}_{p(\theta|z)} \left[ A(z,\theta) \right] - A(z,\theta^*(z)) \tag{139}$$

*where $z$ is fixed, $\theta^*(z) = \mathbb{E}_{p(\theta|z)} [\theta]$ and $A(z,\theta)$ is the partition function of $p(x|z,\theta)$.*

*Proof.* Let $\theta^*(z) = \mathbb{E}_{p(\theta|z)}[\theta]$. Recall that $p(x|z,\theta) = \exp\{\langle t_z(x),\theta\rangle - A(z,\theta)\}$, $q^*(x|z) = p(x|z,\theta^*(z))$ and that the CMFVI objective can be written using an identity from Lemma 4:

$$\mathrm{KL}\left[q(x|z)p(\theta|z)\|p(x,\theta|z)\right] = \mathbb{E}_{q(x|z)}\left[\log q(x|z) - \mathbb{E}_{p(\theta|z)}\left[\log p(x|z,\theta)\right]\right] \quad (140)$$

We can plug $q^*(x|z)$ and $p(x|z,\theta)$ into the identity to get:

$$\mathrm{KL}\left[q^*(x|z)p(\theta|z)\|p(x,\theta|z)\right] \quad (141)$$
$$= \mathbb{E}_{q^*(x|z)}\left[\log q^*(x|z) - \mathbb{E}_{p(\theta|z)}\left[\log p(x|z,\theta)\right]\right] \quad (142)$$
$$= \mathbb{E}_{q^*(x|z)}\left[(\langle t_z(x),\theta^*(z)\rangle - A(z,\theta^*(z))) - \left(\langle t_z(x), \underbrace{\mathbb{E}_{p(\theta|z)}[\theta]}_{\theta^*(z)}\rangle - \mathbb{E}_{p(\theta|z)}[A(z,\theta)]\right)\right] \quad (143)$$
$$= \mathbb{E}_{p(\theta|z)}\left[A(z,\theta)\right] - A(z,\theta^*(z)) \quad (144)$$

$\square$

**Proposition 14** (Forward KL divergence). *The forward KL divergence between $p(x|z)$ and $q^*(x|z)$ is given by:*

$$\mathrm{KL}\left[p(x|z)\|q^*(x|z)\right] = -H_p[x|z] - \langle t^*(z),\theta^*(z)\rangle + A(z,\theta^*(z)) \quad (145)$$

*where $H_p[x|z]$ is the differential entropy of $p(x|z)$, $t^*(z) = \mathbb{E}_{p(x|z)}[t_z(x)]$, $\theta^*(z) = \mathbb{E}_{p(\theta|z)}[\theta]$ and $A(z,\theta)$ is the partition function of $p(x|z,\theta)$.*

*Proof.* This follows from a direct computation:

$$\mathrm{KL}\left[p(x|z)\|q^*(x|z)\right] = -H_p[x|z] - \int p(x|z)\log q^*(x|z)dx \quad (146)$$
$$= -H_p[x|z] - \int p(x|z)\left(\langle t_z(x),\theta^*(z)\rangle - A(z,\theta^*(z))\right)dx \quad (147)$$
$$= -H_p[x|z] - \langle\int p(x|z)t_z(x)dx,\theta^*(z)\rangle + A(z,\theta^*(z)) \quad (148)$$
$$= -H_p[x|z] - \langle t^*(z),\theta^*(z)\rangle + A(z,\theta^*(z)) \quad (149)$$

$\square$

## F.2 Bayes estimator equivariance

We will use the equivariance of the Bayes estimator to linear transformations to show that it is also equivariant to message passing updates when the Gaussian potential functions of the corresponding CRF have covariances that only depend on the node index. This result will allow us to reparameterize the Bayes estimator of the backward messages in terms of the previously computed backward messages, and also in terms of the potential function means themselves. This will be useful for relating the CMFVI time series models we construct back traditional time series models, and also for proving that the autoregressive CMFVI model we construct is an approximation of flow-based generative models for time series.

**Corollary 7** (Commutativity of Bayes estimator with update and marginalize operator). *Let $\phi_{k+1|k}(x_{k+1}|x_k)$ be a Gaussian transition function and let $\phi(x_{k+1}|\eta_{k+1}) := N(x_{k+1}|\mu_{k+1}(y), J_{k+1}^{-1})$ be a Gaussian node potential where $y \sim p(y)$ is an auxilary variable set of variables that only the mean of the potential depends on. Then the Bayes estimator of $\eta_k$ commutes with the update and marginalize operator. That is,*

$$\mathbb{E}_{p(y)}[\eta_k(y)] = \mathbb{E}_{p(y)}[\Phi_{k,k+1}(\eta_{k+1}(y))] = \Phi_{k,k+1}\left(\mathbb{E}_{p(y)}[\eta_{k+1}(y)]\right) \quad (150)$$

*Proof.* We can examine the form of $\Phi_{k,k+1}$ from Corollary 2 to see that $\Phi_{k,k+1}$ is linear with respect to $\mu_{k+1}(y)$. Then the result follows from linearity equivariance of the Bayes estimator. $\square$

 **F.3   CMFVI time series models**

 **Proposition 15** (Naive CMFVI solution). *Let $p(x_{t_{1:N}}|y_{\mathcal{O}})$ be the target distribution. Then the naive*
 *CMFVI solution, denoted by $q^{CRF}(x_{t_{1:N}})$ is the CMFVI approximation of $p(x_{t_{1:N}}|y_{\mathcal{O}})$ and is given*
 *by:*

$$q^{CRF}(x_{t_{1:N}}) \propto \prod_{t_k \in \mathcal{T}} \phi_{t_{k+1}|t_k}(x_{t_{k+1}}|x_{t_k}) \prod_{t_k \in \mathcal{R}} \phi(x_{t_k}|\theta^*_{t_k}(y_{\mathcal{O}})) \tag{151}$$

 *where $\theta^*_{t_k}(y_{\mathcal{O}}) = \mathbb{E}_{p(y_{\mathcal{U}}|y_{\mathcal{O}})}[\theta_{t_k}(y_{\tau_{1:T}})]$ is the Bayes estimator of $\theta_{t_k}$.*

 *Proof.* By expanding $q^*$ using Lemma 5, one finds that the terms of the log likelihood is linear with
 respect to $\theta_{t_k}(y_{\tau_{1:T}})$. Then the result follows from the equivariance of the Bayes estimator to linear
 transformations. $\qquad\square$

 **Proposition 16** (CMFVI transition approximation). *Let $p(x_{t_{1:N}}|y_{\mathcal{O}})$ be the target distribution and*
 *consider its k'th autoregressive factor $p(x_{t_k}|x_{t_{1:k-1}}, y_{\mathcal{O}})$. Then the CMFVI transition approximation*
 *is given by:*

$$q^{transition}(x_{t_k}|x_{t_{1:k-1}}, y_{\mathcal{O}}) \propto \phi_{t_k|t_{k-1}}(x_{t_k}|x_{t_{k-1}})\phi(x_{t_k}|\beta^*_{t_k}(x_{t_{1:k-1}}, y_{\mathcal{O}})) \tag{152}$$

 *where $\beta^*_{t_k}(x_{t_{1:k-1}}, y_{\mathcal{O}}) = \mathbb{E}_{p(y_{\mathcal{U}}|x_{t_{1:k-1}}, y_{\mathcal{O}})}[\beta_{t_k}(y_{\tau_{1:T}})]$ is the Bayes estimate of $\beta_{t_k}(y_{\tau_{1:T}})$, which is*
 *defined using the message passing update operator $\Phi_{t_k,t_{k+1}}$ from Definition 7 as:*

$$\beta_{t_k} = \begin{cases} \Phi_{t_k,t_{k+1}}(\beta_{t_{k+1}}(y_{\tau_{1:T}}) + \theta_{t_{k+1}}(y_{\tau_{1:T}})) & \text{if } t_{k+1} \in \mathcal{R} \\ \Phi_{t_k,t_{k+1}}(\beta_{t_{k+1}}(y_{\tau_{1:T}})) & \text{otherwise} \end{cases} \tag{153}$$

 *Proof.* The transition distribution in the fully observed setting is given by:

$$p(x_{t_k}|x_{t_{1:k-1}}, y_{\tau_{1:T}}) = p(x_{t_k}|x_{t_{k-1}}, y_{\tau_{1:T}}) \tag{154}$$
$$\propto \phi_{t_k|t_{k-1}}(x_{t_k}|x_{t_{k-1}})\phi(x_{t_k}|\beta_{t_k}(y_{\tau_{1:T}})) \tag{155}$$

 If we expand the log likelihood of $p(x_{t_k}|x_{t_{1:k-1}}, y_{\tau_{1:T}})$, we would find that the log likelihood is linear
 with respect to $\beta_{t_k}(y_{\tau_{1:T}})$, and so writing the CMFVI solution using Eq. (136) yields the result. $\quad\square$

 We denote this model by $q^{\text{MSE}}(x_{t_{1:N}}|y_{\mathcal{O}})$.

 **Corollary 8** (MSE Forecaster). *Let $p(x_{t_{1:N}}|y_{\mathcal{O}})$ be the target distribution and suppose the co-*
 *variances of its potentials are constant with respect to y. Then the MSE-CMFVI solution, de-*
 *noted by $q^{MSE}(x_{t_{1:N}})$ is the CMFVI approximation of $p(x_{t_{1:N}}|y_{\mathcal{O}})$ obtained by choosing $(x, z, \theta) =$*
 *$(x_{t_{1:N}}, y_{\mathcal{O}}, \theta(y_{\tau_{1:T}}))$:*

$$q^{MSE}(x_{t_{1:N}}|y_{\mathcal{O}}) \propto \prod_{t_k \in \mathcal{T}} \phi_{t_{k+1}|t_k}(x_{t_{k+1}}|x_{t_k}) \prod_{t_k \in \mathcal{R}} N(x_{t_k}|\mu^*_{t_k}(y_{\mathcal{O}}), \Sigma_{t_k}) \tag{156}$$

 *where $\mu^*_{t_k}(y_{\mathcal{O}}) = \mathbb{E}_{p(y_{\mathcal{U}}|y_{\mathcal{O}})}[\mu_{t_k}(y_{\tau_{1:T}})]$ is the Bayes estimate of $\mu_{t_k}$, and $\phi(x_{t_k}|\theta_{t_k}(y_{\tau_{1:T}})) =$*
 *$N(x_{t_k}|\mu^*_{t_k}(y_{\tau_{1:T}}), \Sigma_{t_k})$.*

 See Appendix F.3 for a proof.

 **Definition 8** (Autoregressive CMFVI solution). *Let $p(x_{t_{1:N}}|y_{\mathcal{O}})$ be the target distribution. Then the*
 *autoregressive CMFVI solution, denoted by $q^{AR}(x_{t_{1:N}})$ is the CMFVI approximation of $p(x_{t_{1:N}}|y_{\mathcal{O}})$*
 *and is given by:*

$$q^{AR}(x_{t_{1:N}}) \propto p(x_{t_1}|y_{\mathcal{O}}) \prod_{t_k \in \mathcal{T}} q^{transition}(x_{t_k}|x_{t_{1:k-1}}, y_{\mathcal{O}}) \tag{157}$$

 *where $q^{transition}(x_{t_k}|x_{t_{1:k-1}}, y_{\mathcal{O}})$ is the CMFVI transition approximation given by Proposition 6.*

 **Corollary 9** (MSE Forecaster). *Let $p(x_{t_{1:N}}|y_{\mathcal{O}})$ be the target distribution and suppose the covari-*
 *ances of its potentials are constant with respect to y. Then the MSE-CMFVI solution, denoted by*
 *$q^{MSE}(x_{t_{1:N}})$ is the CMFVI approximation of $p(x_{t_{1:N}}|y_{\mathcal{O}})$ and is given by:*

$$q^{MSE}(x_{t_{1:N}}) \propto \prod_{t_k \in \mathcal{T}} \phi_{t_{k+1}|t_k}(x_{t_{k+1}}|x_{t_k}) \prod_{t_k \in \mathcal{R}} N(x_{t_k}|\mu^*_{t_k}(y_{\mathcal{O}}), \Sigma_{t_k}) \tag{158}$$

 *where $\mu^*_{t_k}(y_{\mathcal{O}}) = \mathbb{E}_{p(y_{\mathcal{U}}|y_{\mathcal{O}})}[\mu_{t_k}(y_{\tau_{1:T}})]$ is the Bayes estimate of $\mu_{t_k}$.*

*Proof.* This follows from the fact that the potentials are constant with respect to $y$ and the linear equivariance of the Bayes estimator. $\qquad\square$

**Corollary 10** (Autoregressive MSE Forecaster). *Let $p(x_{t_{1:N}}|y_\mathcal{O})$ be the target distribution and suppose the covariances of its potentials are constant with respect to $y$. Then the autoregressive MSE-CMFVI solution, denoted by $q^{\text{AR-MSE}}(x_{t_{1:N}})$ is the CMFVI approximation of $p(x_{t_{1:N}}|y_\mathcal{O})$ and is given by:*

$$q^{\text{AR-MSE}}(x_{t_{1:N}}) \propto p(x_{t_1}|y_\mathcal{O}) \prod_{t_k \in \mathcal{T}} \phi_{t_k|t_{k-1}}(x_{t_k}|x_{t_{k-1}}) \prod_{t_k \in \mathcal{R}} N\left(x_{t_k} \mid \left(\mu_{t_k}^\beta\right)^* (x_{t_{1:k}}, y_\mathcal{O}), \Sigma_{t_k}^\beta\right) \quad (159)$$

*where $\left(\mu_{t_k}^\beta\right)^* (x_{t_{1:k}}, y_\mathcal{O}) = \mathbb{E}_{p(y_\mathcal{U}|x_{t_{1:k}}, y_\mathcal{O})}\left[\mu_{t_k}^\beta(y_{\tau_{1:T}})\right]$ is the Bayes estimate of $\mu_{t_k}^\beta$ and $\Sigma_{t_k}^\beta$ is the covariance of the backward message of $p(x_{t_{1:N}}|y_{\tau_{1:T}})$.*

*Proof.* This follows from the fact that the potentials are constant with respect to $y$ and the linear equivariance of the Bayes estimator. $\qquad\square$

**Definition 9** (Continuous extension of AR-MSE model). *Let $q^{\text{AR}}$ be the autoregressive CMFVI solution and consider the setting where the potential functions of $p(x_{t_{1:N}}|y_{\tau_{1:T}})$ have covariances that do not depend on $y$. Then the continuous extension of $q^{\text{AR}}$ is given by the following piecewise linear SDE:*

$$dx_t = (F_t x_t + L_t L_t^T \nabla \log \phi(x_t|\beta_t^*(x_{t_{1:k}}, y_\mathcal{O})))dt + L_t dW_t, \quad (160)$$

$$\text{where } \beta_t^*(x_{t_{1:k}}, y_\mathcal{O}) = \mathbb{E}_{p(y_\mathcal{U}|x_{t_{1:k}}, y_\mathcal{O})}\left[\beta_t(y_{\tau_{1:T}})\right], \quad \text{and } t \in (t_k, t_{k+1}) \quad (161)$$

*where $\beta_t^*(x_{t_{1:k}}, y_\mathcal{O})$ is the Bayes estimator of $\beta_t(y_{\tau_{1:T}}) = \Phi_{t,t_{k+1}}(\beta_{t_{k+1}}(y_{\tau_{1:T}}))$.*

*Proof.* We just need to verify that this piecewise linear SDE has the same joint distribution as $q^{\text{AR}}$ on $t_{1:N}$. To do this, we can just check that each of the linear SDEs that are defined on the intervals $(t_k, t_{k+1})$ have the same joint distribution as $q^{\text{transition}}(x_{t_k}|x_{t_{1:k-1}}, y_\mathcal{O})$ from Proposition 6. This is true by construction TODO: add proof. $\qquad\square$

# G    Flow-based generative models proofs

In this section we provide basic results about Bayes estimation for generalized linear stochastic interpolants. Let $dx_t = F_t x_t dt + L_t dW_t$ be the base linear SDE and let the distribution of random draws, at times $t_{1:N}$, be denoted by $p(x_{t_{1:N}}|c)$. Let $p(x_{t_{1:N}}|\theta, c)$ be its conditional distribution given parameters $\theta$ that are only available during training time and some extra conditioning information $c$ that is avilable at both training and test time, and suppose that $p(\theta|c)$ is the (unknown) distribution of $\theta$ given $c$. The goal of the techniques in this section (and FBGMs in general), is to construct, and learn, the distribution of $p(x_{t_{1:N}}|c)$, which is the distribution needed to generate samples of $x_{t_{1:N}}$ when we do not have access to the parameters $\theta$. At a high level, FBGMs offer different inference algroithms for this task. In this section, we will derive three of these inference algorithms.

## G.1    Score function for FBGMs

**Proposition 17** (Score function for FBGMs). *Suppose that $p(\theta|c)$ is a probability distribution over $\theta$ given some extra conditioning information $c$ and $p(x_t|\theta, c)$ is the marignal distribution of a generalized linear stochastic interpolant whose base linear SDE is given by $dx_t = F_t x_t dt + L_t dW_t$. Then the score function of $p(x_t|c)$ is given by:*

$$\nabla \log p(x_t|c) = \nabla \log \phi(x_t|\alpha_t^*(x_t, \theta, c) + \beta_t^*(x_t, \theta, c)) \quad (162)$$

*where $\alpha_t^*(x_t, \theta, c) = \mathbb{E}_{p(\theta|x_t, c)}\left[\alpha_t(\theta, c)\right]$ and $\beta_t^*(x_t, \theta, c) = \mathbb{E}_{p(\theta|x_t, c)}\left[\beta_t(\theta, c)\right]$ are Bayes estimators of the forward and backward messages to time $t$ using $x_t$ respectively.*

*Proof.* A straightforward calculation will lead to the desired result.

$$\nabla \log p(x_t|c) = \frac{1}{p(x_t|c)} \nabla p(x_t|c) \quad (163)$$

$$= \frac{1}{p(x_t|c)} \nabla \int p(\theta|c) p(x_t|\theta, c) d\theta \tag{164}$$

$$= \frac{1}{p(x_t|c)} \int p(\theta|c) \nabla p(x_t|\theta, c) d\theta \tag{165}$$

$$= \int \frac{p(\theta|c) p(x_t|\theta, c)}{p(x_t|c)} \nabla \log p(x_t|\theta, c) d\theta \tag{166}$$

$$= \mathbb{E}_{p(\theta|x_t, c)} \left[ \nabla \log p(x_t|\theta, c) \right] \tag{167}$$

$$= \mathbb{E}_{p(\theta|x_t, c)} \left[ \nabla \log \phi(x_t|\alpha_t(\theta, c) + \beta_t(\theta, c)) \right] \quad \because Lemma\ 2 \tag{168}$$

$$= \nabla \log \phi(x_t|\alpha_t^*(x_t, \theta, c) + \beta_t^*(x_t, \theta, c)) \quad \because Eq.\ (21) \tag{169}$$

□

## G.2   General form of Markovian projection SDE

**Lemma 6** (General form of Markovian projection SDE). *Suppose that $p(\theta|c)$ is a probability distribution over $\theta$ given some extra conditioning information $c$ and $p(x_t|\theta, c)$ is the marginal distribution of a generalized linear stochastic interpolant whose base linear SDE is given by $dx_t = F_t x_t dt + L_t dW_t$. Then the Markovian projection SDE is given by:*

$$dx_t = (F_t x_t + L_t L_t^T \nabla \log \phi(x_t|\beta_t^*(x_t, \theta, c))) dt + L_t dW_t \tag{170}$$

*where $\beta_t^*(x_t, \theta, c) = \mathbb{E}_{p(\theta|x_t, c)} \left[ \beta_t(\theta, c) \right]$ is the Bayes estimate of the backward message to time $t$ using $x_t$.*

*Proof.* The Markovian projection SDE is the SDE whose marginal distribution evolves in time in the same way that $p(x_t|c)$ evolves in time, and so our proof strategy will follow the same strategy as [Lipman et al., 2023, Theorem 1] where we take the time derivative of $p(x_t|c)$ and recognize the form of the SDE.

First, recall that the Fokker-Planck equation [Särkkä and Solin, 2019, Øksendal and Øksendal, 2003] relates an SDE to the time derivative of its marginal distribution. Let $p(x_t|\theta, c)$ be the marginal distribution of the generalized linear stochastic interpolant and recall that its corresponding SDE is given by $dx_t = (F_t x_t + L_t L_t^T \nabla \log \phi(x_t|\beta_t(\theta, c))) dt + L_t dW_t$ (see Proposition 4). Then the Fokker-Planck equation for this SDE is given by:

$$\frac{\partial p(x_t|\theta, c)}{\partial t} = -\text{Div}(p(x_t|\theta, c)(F_t x_t + L_t L_t^T \nabla \log \phi(x_t|\beta_t(\theta, c)))) + \frac{1}{2} L_t L_t^T \text{Div}(\nabla p(x_t|\theta, c)) \tag{171}$$

$L_t L_t^T$ appears outside the divergence operator because it does not depend on $x_t$. Next, we can directly take the time derivative of $p(x_t|c)$ and recognize the form of the corresponding SDE.

$$\frac{\partial p(x_t|c)}{\partial t} = \mathbb{E}_{p(\theta|c)} \left[ \frac{\partial p(x_t|\theta, c)}{\partial t} \right] \tag{172}$$

$$= \mathbb{E}_{p(\theta|c)} \left[ -\text{Div}(p(x_t|\theta, c)(F_t x_t + L_t L_t^T \nabla \log \phi(x_t|\beta_t(\theta, c)))) + \frac{1}{2} L_t L_t^T \text{Div}(\nabla p(x_t|\theta, c)) \right] \tag{173}$$

$$= \mathbb{E}_{p(\theta|c)} \left[ -\text{Div}(p(x_t|\theta, c) F_t x_t) \right] \quad \text{(A)} \tag{174}$$

$$+ \mathbb{E}_{p(\theta|c)} \left[ -\text{Div}(p(x_t|\theta, c) L_t L_t^T \nabla \log \phi(x_t|\beta_t(\theta, c))) \right] \quad \text{(B)} \tag{175}$$

$$+ \mathbb{E}_{p(\theta|c)} \left[ \frac{1}{2} L_t L_t^T \text{Div}(\nabla p(x_t|\theta, c)) \right] \quad \text{(C)} \tag{176}$$

Since all of the divergence and gradient operators depend only on $x_t$, we can pass the expectation through these terms. We can simplify each terms as follows:

**(A)**

$$\mathbb{E}_{p(\theta|c)} \left[ -\text{Div}(p(x_t|\theta, c) F_t x_t) \right] = -\text{Div}(p(x_t|c) F_t x_t) \tag{177}$$

**(B)**

$$\mathbb{E}_{p(\theta|c)}\left[-\text{Div}(p(x_t|\theta,c)L_tL_t^T\nabla\log\phi(x_t|\beta_t(\theta,c)))\right] = -\text{Div}(\int p(\theta|c)p(x_t|\theta,c)L_tL_t^T\nabla\log\phi(x_t|\beta_t(\theta,c))d\theta)$$
(178)

$$= -\text{Div}(\int p(\theta|x_t,c)p(x_t|c)L_tL_t^T\nabla\log\phi(x_t|\beta_t(\theta,c))d\theta)$$
(179)

$$= -\text{Div}(p(x_t|c)L_tL_t^T\mathbb{E}_{p(\theta|x_t,c)}\left[\nabla\log\phi(x_t|\beta_t(\theta,c))\right])$$
(180)

**(C)**

$$\mathbb{E}_{p(\theta|c)}\left[\frac{1}{2}L_tL_t^T\text{Div}(\nabla p(x_t|\theta,c))\right] = \frac{1}{2}L_tL_t^T\text{Div}(\nabla\mathbb{E}_{p(\theta|c)}\left[p(x_t|\theta,c)\right])$$
(181)

$$= \frac{1}{2}L_tL_t^T\text{Div}(\nabla p(x_t|c))$$
(182)

Putting these terms back together, we get:

$$\frac{\partial p(x_t|c)}{\partial t} = -\text{Div}(p(x_t|c)\underbrace{\left(F_tx_t + L_tL_t^T\mathbb{E}_{p(\theta|x_t,c)}\left[\nabla\log\phi(x_t|\beta_t(\theta,c))\right]\right)}_{\text{recognize as drift term in Fokker-Planck equation}}) + \frac{1}{2}L_tL_t^T\text{Div}(\nabla p(x_t|c))$$
(183)

We can see that the form of the Markovian projection SDE is given by:

$$dx_t = \left(F_tx_t + L_tL_t^T\mathbb{E}_{p(\theta|x_t,c)}\left[\nabla\log\phi(x_t|\beta_t(\theta,c))\right]\right)dt + L_tdW_t$$
(184)

Lastly because $\phi(x_t|\beta_t(\theta,c))$ is a Gaussian distribution with natural parameters $\beta_t(\theta,c)$, its pdf is given by:

$$\phi(x_t|\beta_t(\theta,c)) = \exp\{\langle t_c(x_t),\beta_t(\theta,c)\rangle - A(c,\theta)\}$$
(185)
(186)

where $t_c(x_t)$ is the sufficient statistic of the Gaussian distribution and $A(c,\theta)$ is the log partition function. From this form, we can immediately see that the expectation around the score function passes through to the natural parameters:

$$\mathbb{E}_{p(\theta|x_t,c)}\left[\nabla\log\phi(x_t|\beta_t(\theta,c))\right] = \langle\nabla t_c(x_t),\mathbb{E}_{p(\theta|x_t,c)}\left[\beta_t(\theta,c)\right]\rangle$$
(187)

If we let $\beta_t^*(x_t,\theta,c) = \mathbb{E}_{p(\theta|x_t,c)}\left[\beta_t(\theta,c)\right]$ and stop the gradient with respect to $x_t$ through $\beta_t^*$, then we recover the desired result. $\square$

**Proposition 18** (Neural latent SDE). *Let $p(x_{t_{1:N}},y_{1:T})$ be the joint distribution defined in Definition 2 and suppose that $\mathbf{y} = (y_{\mathcal{O}},y_{\mathcal{U}})$, where $\mathcal{O}$ and $\mathcal{U}$ are the times at which sequences are observed and unobserved, respectively. Then the neural latent SDE is the following piecewise SDE defined on the intervals $(t_k,t_{k+1})$ for $k = 1,\ldots,N$:*

$$dx_t = (F_tx_t + L_tL_t^T\nabla\log\phi(x_t|\beta_t^*(x_t,x_{t_{1:k}},y_{\mathcal{O}})))dt + L_tdW_t,$$
(188)

$$\text{where } \beta_t^*(x_t,x_{t_{1:k}},y_{\mathcal{O}}) = \mathbb{E}_{p(y_{\mathcal{U}}|x_t,x_{t_{1:k}},y_{\mathcal{O}})}\left[\beta_t(y_{1:T})\right], \text{ and } t \in (t_k,t_{k+1})$$
(189)

$\beta_t^*(x_t,x_{t_{1:k}},y_{\mathcal{O}})$ *is the Bayes estimator of $\beta_t$ using the current state $x_t$.*

*Proof.* The result follows directly from Lemma 6 by choosing $\theta = y_{\mathcal{U}}$ and $c = x_{t_{1:k}}$. $\square$

## G.3 General form of Markovian projection ODE

**Lemma 7** (General form of Markovian projection ODE). *Suppose that $p(\theta|c)$ is a probability distribution over $\theta$ given some extra conditioning information $c$ and $p(x_t|\theta,c)$ is the marginal distribution of a generalized linear stochastic interpolant whose base linear SDE is given by $dx_t = F_tx_tdt + L_tdW_t$. Then the Markovian projection ODE is defined as the probability flow ODE of the Markovian projection SDE and is given by:*

$$\frac{dx_t}{dt} = F_tx_t + \frac{1}{2}L_tL_t^T\left(\nabla\log\phi(x_t|\beta_t^*(x_t,\theta,c)) - \nabla\log\phi(x_t|\alpha_t^*(x_t,\theta,c))\right)$$
(190)

*where $\beta_t^*(x_t,\theta,c) = \mathbb{E}_{p(\theta|x_t,c)}\left[\beta_t(\theta,c)\right]$ and $\alpha_t^*(x_t,\theta,c) = \mathbb{E}_{p(\theta|x_t,c)}\left[\alpha_t(\theta,c)\right]$ are Bayes estimators of the forward and backward messages to time $t$ using $x_t$ respectively.*

*Proof.* Recall that the definition of the probability flow ODE of an SDE of the form $dx_t = u_t(x_t)dt + L_t dW_t$ is given by [Song et al., 2021]:

$$\frac{dx_t}{dt} = u_t(x_t) - \frac{1}{2}L_t L_t^T \nabla \log p(x_t|c) \tag{191}$$

Plugging in drift of the Markovian projection SDE in Lemma 6, and the score function of $p(x_t|c)$ in Proposition 17, we get the desired result. $\square$

# H   Message Passing Implementation Details

We devise a careful implementation of message passing to ensure numerical stability. There are many different ways to implement message passing. For example, [Särkkä et al., 2006] parameterizes the potentials in the standard form of Gaussians and uses Kalman filtering [Kalman, 1960] to obtain the forward messages and does not directly compute the backward messages, but instead uses the Rauch-Tung-Striebel smoother [Rauch et al., 1965] to blend the forward and backward message computations to obtain the smoothed potentials. Alternatively, [Fox, 2009, Johnson and Linderman, 2015] utilize a natural parameterization of the potentials in order to have simple message passing updates. Our implementation requires that we can express both total uncertainty, and total certainty, in a variable in order to be able to work with incomplete, or missing data, and to condition exactly on variables. To do this, we adopt a mixed parametrization that contains the mean of the Gaussian and precision matrix so that we can express total uncertainty using a precision matrix of $0$ and total certainty in the mean value by using a symbolic infinity. We also use symbolic zeros to mitigate accumulation of errors when perform message passing on long chains of latent variables without any evidence.

## H.1   Numerical stability considerations

Before we look at the implementation details, we will look at what considerations we need to make for the implementation of these operations in a numerically stable way. Recall that the transition distribution of an LTI-SDE is given by

$$\phi(x_{t+s}|x_t) = N(x_{t+s}|A_s x_t, \Sigma_s) \tag{192}$$

where

$$\begin{bmatrix} A_s & \Sigma_s A_s^{-T} \\ 0 & A_s^{-T} \end{bmatrix} := \exp\{\begin{bmatrix} F & LL^T \\ 0 & -F^T \end{bmatrix} s\} \tag{193}$$

and that potential functions can be written in natural or standard form as:

$$\phi(x) = \exp\{-\frac{1}{2}x^T J x + x^T h - \log Z\} \tag{194}$$

$$= \exp\{-\frac{1}{2}x^T \Sigma^{-1} x + x^T \Sigma^{-1} \mu - \log Z\} \tag{195}$$

where $\Sigma = J^{-1}$ and $\mu = J^{-1}h$. We assume that the time intervals between consecutive variables are bounded and nonzero so that $\Sigma_s$, $A_s$, and $A_s^{-T}$ are numerically stable. We also assume that the covariance matrices that the user specifies for the node potentials, e.g. $\Sigma$ or $J$, are well conditioned. We do not assume that $\Sigma_s^{-1}$, $\Sigma^{-1}$ nor $J^{-1}$ are well conditioned. These assumptions are made to accomodate operations that a user might perform in practice. For example, a user may choose to express $0$ certainty in a variable by setting $\Sigma \to \infty$ or $J = 0$ and can choose to express $0$ uncertainty by setting $\Sigma = 0$ or $J \to \infty$. Furthermore, if a user chooses to discretize an SDE at points where $s$ is small, or even exactly $0$, then $\Sigma_s$ is close to $0$ and so $\Sigma_s^{-1}$ can be very large. To account for these considerations, we use symbolic computation to represent matrices that are $0$ or $\infty$ as needed. Furthermore, we use three different parameterizations of the Gaussian to ensure that we can handle all cases. We use the **standard** parameterization, $(\mu, \Sigma)$, **natural** parameterization [3], $(J = \Sigma^{-1}, h = \Sigma^{-1}\mu)$, and **mixed** parameterization $(J = \Sigma^{-1}, \mu)$. For brevity, we will not include the updates for the normalizing constant $\log Z$ in our pseudocode.

---

[3]The true natural parameters are scaled by $-\frac{1}{2}$

 **H.2 Message passing pseudocode**

 In Appendix D we identified the key operations that are needed to perform variable elimination in the
 sequential and parallel settings (see Appendices D.1 and D.2). These operations are:

     1. An "add" operation adds the parameters of two potential functions together (code in Ap-
        pendix H.3).

     2. An "update" operation that absorbs a potential function into a transition function (defined in
        Definition 5 and code in Appendix H.3).

     3. A "marginalize" operation that marginalizes out a variable from a Gaussian joint distribution.
        In practice, we fuse this with the "update" operation (code in Appendix H.3).

     4. A "reverse" operation that reverses the direction of a transition (code in Appendix H.3).

     5. A "chain" operation that chains two transition functions (defined in Eq. (40) and code in
        Appendix H.3).

 In Appendix H.3, Appendix H.3, Appendix H.3, and Appendix H.3 we provide pseudocode for
 message passing that involves these operations.

 **H.3 Update rules**

Now we provide pseudocode for the update rules.

---
**Algorithm 1** `Add`

---
    1. Require: potential functions $\phi_1$ and $\phi_2$

    2. $(J_1, h_1) = \texttt{to\_natural}(\phi_1)$

    3. $(J_2, h_2) = \texttt{to\_natural}(\phi_2)$

    4. Return $\texttt{from\_natural}((J_1 + J_2, h_1 + h_2))$

---

---
**Algorithm 2** `Update`

---
    1. Require: potential function $\phi$ and transition $\phi_{k+1|k}$

    2. $(J, \mu) = \texttt{to\_mixed}(\phi)$

    3. $(A, u, \Sigma) = \phi_{k+1|k}$

    4. $R = J(I + \Sigma J)^{-1}$

    5. $S = \Sigma R$

    6. $T = I - S$

    7. $\bar{\phi}_{k+1|k} = (TA, Tu + S\mu, T\Sigma)$

    8. $\bar{\phi} = \texttt{from\_mixed}((A^T R^T A, A^{-1}(\mu - u)))$

    9. $\Psi_{k+1,k} = (\bar{\phi}_{k+1|k}, \bar{\phi})$

    10. Return $\Psi_{k+1,k}$

---

---
**Algorithm 3** `Update and marginalize`

---
    1. Require: potential function $\phi$ and transition $\phi_{k+1|k}$

    2. $(\_, \bar{\phi}) = \texttt{Update}(\phi, \phi_{k+1|k})$

    3. Return $\bar{\phi}$

---

**Algorithm 4** `Reverse`

1. Require: transition $\phi_{k+1|k}$
2. $(A, u, \Sigma) = \phi_{k+1|k}$
3. $\bar{A} = A^{-1}$
4. $\bar{u} = -A^{-1}u$
5. $\bar{\Sigma} = A^{-1}\Sigma A^{-T}$
6. Return $(\bar{A}, \bar{u}, \bar{\Sigma})$

**Algorithm 5** `Chain`

1. Require: transition functions $\phi_{k|k-1}$ and $\phi_{k+1|k}$
2. $A_k, u_k, \Sigma_k = \phi_{k+1|k}$
3. $A_{k-1}, u_{k-1}, \Sigma_{k-1} = \phi_{k|k-1}$
4. $A = A_k A_{k-1}$
5. $u = A_k u_{k-1} + u_k$
6. $\Sigma = \Sigma_k + A_k \Sigma_{k-1} A_k^T$
7. Return $(A, u, \Sigma)$

**Algorithm 6** `BackwardMessagePassing`

1. Require $(\phi_{2|1}, \ldots, \phi_{N|N-1})$ and $(\phi_1, \ldots, \phi_N)$
2. Initialize $\beta_N = 0$
3. For $k = N, \ldots, 2$:
   (a) $\Psi_{k,k-1} = \texttt{Update}(\phi_{k|k-1}, \phi_k + \beta_k)$
   (b) $\beta_{k-1} = \texttt{Marginalize}(\Psi_{k,k-1})$
4. Return $(\beta_1, \ldots, \beta_N)$

**Algorithm 7** `ParallelBackwardMessagePassing`

1. Require $(\phi_{2|1}, \ldots, \phi_{N|N-1})$ and $(\phi_1, \ldots, \phi_N)$
2. In parallel, for $k = N, \ldots, 2$:
   (a) $\Psi_{k,k-1} = \texttt{Update}(\phi_{k|k-1}, \phi_k)$
3. $(\Psi_{1:N}, \ldots, \Psi_{N-1:N}) = \texttt{AssociativeScan}(\texttt{Chain}, \Psi_{2,1}, \ldots, \Psi_{N,N-1})$
4. In parallel, for $k = N - 1, \ldots, 1$:
   (a) $\beta_k = \texttt{Marginalize}(\Psi_{k:N})$
5. $\beta_N = 0$
6. Return $(\beta_1, \ldots, \beta_N)$

---

**Algorithm 8** `ForwardMessagePassing`

---

1. Require $(\phi_{2|1}, \ldots, \phi_{N|N-1}), (\phi_1, \ldots, \phi_N)$ and `use_parallel`
2. For $k = 1, \ldots, N-1$:
    (a) $\phi_{k|k+1} = \texttt{Reverse}(\phi_{k+1|k})$
3. If `use_parallel`:
    (a) `MessagePassing = ParallelBackwardMessagePassing`
4. Else:
    (a) `MessagePassing = BackwardMessagePassing`
5. $(\alpha_N, \ldots, \alpha_1) = \texttt{MessagePassing}((\phi_{N-1|N}, \ldots, \phi_{1|2}), (\phi_N, \ldots, \phi_1))$
6. Return $(\alpha_1, \ldots, \alpha_N)$

---

**Algorithm 9** `AssociativeScan` (Even number of elements only)

---

1. Require: operator $\oplus$, elements $(t_1, t_2, \ldots, t_n)$ where $n$ is a power of 2
2. If $n == 1$:
    (a) Return $t_1$
3. In parallel, for $k = 1, \ldots, n/2$:
    (a) $p_k = t_{2k-1} \oplus t_{2k}$
4. $(r_2, r_4, \ldots, r_n) = \texttt{AssociativeScan}(\oplus, (p_1, p_2, \ldots, p_{n/2}))$
5. In parallel, for $k = 1, \ldots, n/2 - 1$:
    (a) $r_{2k+1} = r_{2k} \oplus t_{2k+1}$
6. $r_1 = t_1$
7. Return $(r_1, r_2, \ldots, r_n)$

---

## I Dataset details

We used two synthetic datasets and five real-world datasets for our experiments - a synthetic noisy double pendulum and synthetic sine wave datasets, and real world datasets for modeling stocks, energy, etth, mujoco, and fmri datasets. For all of our experiments, we use an 80/10/10 split for the training, validation, and test sets. We adopted two different approaches to generate these splits, one for then the dataset only containd a single time series, and one for when the dataset containd multiple time series. For datasets that only contain a single time series, such as the noisy double pendulum, stocks, etth and fmri datasets, we split our data into training, validation, and test sets by splitting the series into three contiguous segments for the training, validation, and test sets respectively, using the 80/10/10 split, and then construct windowed batches of a fixed length for each of the training, validation, and test sets.

## J Model implementation details

### J.1 Neural network architecture and training details

To ensure a fair comparison, we use nearly the exact same neural network architectures and training procedures for all of the models. The architecture that we use is an encoder-decoder transformer architecture where each transformer has 10 layers, 32 heads and a hidden dimension of 128. In between each transformer layer we use a Wavenet convolution block that has 256 channels and uses a kernel size of 4. The observed sequence of variables is passed through the encoder and then used to condition the decoder as it processes the currently generated sequence. We did not do extensive architecture tuning and chose this model early on because it performed well enough for our experiments. We incorporated information about the times in each series by constructing

a feature vector for each scalar time and concatenating it with the observed sequence of variables before passing the contatenation to the transformer. For the models that needed to be autoregressive, we used causal convolutions and causal attention masks to ensure that the Jacobian matrix of the model was lower triangular. See our code for full details.

Each of our models were trained on a single 2080ti GPU using a learning rate of $10^{-4}$ using the adamw optimizer, linear warmup of 1000 steps, and an effective batch size of 256 (we used a batch size of 64 and 4 gradient accumulation steps). For each experiment, we used 5 random seeds to initialize the model parameters and to split the data into training, validation, and test sets using an 80/10/10 split. We evaluated the objective function on the entire validation set every 1000 gradient updates and stopped training when the value of the objective function over the entire validation set stopped improving for 5 evaluations. We normalized the elements of each series by subtracting the mean and dividing by the standard deviation of the first, observed variable in the series to ensure that the elements of each series were on a similar scale.

## J.2  Model details

We implemented 8 different models, of which 6 are latent space forecasters and 2 are observation space forecasters. The baseline, observation space models, were trained to model $p(\mathbf{y}_{k+1:N}|\mathbf{y}_{1:k})$ while the latent space models were trained to model $p(\mathbf{x}_{1:N}|\mathbf{y}_{1:k})$. Of the latent space forecasters, 4 are CMFVI based models and while the last 2 are the same baseline models that we used for the observation space models, just trained on the latent process instead of the observed process.

1. Baselines probabilistic forecasters (Trained to approximate $p(\mathbf{y}_{k+1:N}|\mathbf{y}_{1:k})$):

   (a) Conditional Gaussian autoregressive model

   (b) Diffusion model

2. Latent probabilistic forecasters (Trained to approximate $p(\mathbf{x}_{1:N}|\mathbf{y}_{1:k})$):

   (a) CMFVI models:
      i. MSE forecaster
      ii. Autoregressive MSE forecaster
      iii. Neural ODE
      iv. Neural SDE

   (b) Conditional gaussian autoregressive

   (c) Diffusion model

The encoder networks in each model accept as input $\mathbf{y}_{1:k}$ and output a context embedding that is used to condition the decoder. The decoder accepts as input a sequence of variables that are currently being generated and outputs a sequence of different quantities whose interpretation depends on the model. Next, we will describe each of the models that we implemented, what their decoder outputs are, what their training objective is, and how they generate samples.

**Conditional Gaussian autoregressive model**  The Gaussian conditional chains parameterize the distribution of the next variable in the sequence as a Gaussian distribution. The decoder transformer network outputs the mean and covariance of the next distribution for the entire sequence of generated variables at once. Since the decoder is autoregressive, the mean and covariance of the next distribution is found at the same position as the most recently generated variable. For the latent space model, the first variable is sampled from a CRF, of the same kind used to construct the latent process, that is conditioned on the observed variables. The model is trained to maximize the log likelihood of the unobserved sequence given the observed sequence.

**Diffusion model**  The diffusion model is trained using flow-matching [Lipman et al., 2023] using a brownian bridge between a Gaussian random variable and the sequence of unobserved variables. This model is effectively the same as standard diffusion models for images, but applied to a flattened time series vector. The decoder transformer network outputs the vector field of the probability flow ODE that is used to simulate the process. Samples are generated by passing a sequence of Gaussian random variables of the same size as $\mathbf{y}_{k+1:N}$ to an ODE solver that uses the vector field output by the decoder to simulate the process.

**MSE forecaster**  The MSE forecaster predicts the mean of the potential functions of the CRF used to construct the latent process. This model is trained to minimize the mean squared error between the predicted mean of each potential function, and the mean of the potential function of the target process. To generate samples from this model, we use the input $\mathbf{y}_{1:k}$ to generate the means of the CRF potentials for the entire sequence of generated variables. We then sample from the CRF defined by these potentials to get a sample from this model.

**Autoregressive MSE forecaster**  This model is also a conditional Gaussian autoregressive model, except that the model only parameterizes the mean of each transition distribution, and not the covariance, because, as mentioned in (REF), when the covariance matrices of the potential functions do not depend on the values of $\mathbf{y}$, then the covariance matrices are known analytically using Kalman smoothing. To train this model, we minimize the mean squared error between the means of the true transition distributions (using the entire observed sequence), $p(\mathbf{x}_{i+1}|\mathbf{x}_i, \mathbf{y}_{1:N})$, and the mean predicted by our model for $q(\mathbf{x}_{i+1}|\mathbf{x}_i, \mathbf{y}_{1:k})$. We generate samples from this model using the same procedure as the one for the conditional Gaussian autoregressive model defined above.

**Neural ODE/SDE**  We designed a novel parameterization of neural process models based on flow-based generative models in order to be able to use the same autoregressive transformer architecture as the other models, and also to make these scalable during training. Recall that a single step of training a flow-based generative model requires constructing a stochastic bridge between samples from a source and target distribution, sampling a random time in between the source and target time, sampling from the stochastic bridge at this time and then computing the probability flow ODE vector (or drift) of the bridge at this time. To extend this to time series, we must be able to perform this procedure for every pair of consecutive time points in a time series. To this end, we construct our transformer decoder to take as input the latent sequence that we are generating at the fixed set of times $\mathcal{T} := \{t_1, \ldots, t_N\}$ and also elements of the latent sequence at (uniformly) random times inbetween these times, compute both the predicted and true control (either probability flow ODE vector or drift vector) at both the original and new times, and then return the mean squared error between the two.

More formally, at training time suppose that we uniformly sample times in between the times in $\mathcal{T}$ as $\tau_i \sim \mathcal{U}(t_i, t_{i+1})$ for $i = 1, \ldots, N - 1$. Then we can sample from the stochastic bridge at these times to get a sample from the model, $\mathbf{x}_{\mathcal{T}+\tau} \sim p(\mathbf{x}_{\mathcal{T}+\tau}|\mathbf{y}_{1:N})$, where $\mathbf{x}_{\mathcal{T}+\tau} := (x_{t_1}, x_{\tau_1}, x_{t_2}, x_{\tau_2}, \ldots, x_{\tau_{N-1}}, x_{t_N})$. Our decoder transformer network takes as input $\mathbf{x}_{\mathcal{T}+\tau}$ and the embedding of $\mathbf{y}_{1:k}$ from the encoder and outputs the probability flow ODE vector (if we are training a neural ODE) or the drift vector (if we are training a neural SDE) at the times $\mathcal{T} + \tau$. Our conditioned linear SDE library allows us to efficiently sample from $p(\mathbf{x}_{\mathcal{T}+\tau}|\mathbf{y}_{1:N})$, as well as compute the target control vector for the samples. We then compute the mean squared error between the predicted control vector and the target control vector to get our loss function. Since we ensure that our decoder network is autoregressive, we are able to compute the loss for the drift for the entire sequence at once, rather than having to compute for a single time step as is the case in existing implementations of these kinds of models (CITE).

Our sample generation procedure simulates and ODE/SDE where the control vector at time $t$ is given by the $k$'th element of the decoder output, where $t \in (t_k, t_{k+1})$. To begin, we first sample an initial point from $p_{\mathrm{CRF}}(x_{t_0}|\mathbf{y}_{1:k})$. Note that this distribution is not equal to the target $p(x_{t_0}|\mathbf{y}_{1:k})$, but is a reasonable approximation if $k$ is reasonably large. Then we sample a set of times, $\tau$, in between the times in $\mathcal{T}$, like we do during training, to hold the intermediate variables that we store in order to feed the neural network an input that looks similar to the one used during training. The sampling procedure can be broken down into a sequence of $k$ steps, where at step $k \in [0, N)$, we simulate the variable $x_{t_k}$ forward in time from time $t = t_k, t_{k+1}$ to predict the next element of the sequence, $x_{t_{k+1}}$. At the first step, we initialize the buffer of $2N - 1$ elements $(x_{t_0}, 0, \ldots, 0)$. Then for each step $k \in [0, N)$, we simulate the variable $x_{t_k}$ forward in time from time $t = t_{k-1}, t_k$ to predict the next element of the sequence, $x_{t_k}$. The control of this simulation process is computed by passing the current buffer of variables to the decoder network. During simulation, we record the value of the process at the time, $\tau_k$, so that at the end of step $k$, we update the buffer to include both $x_{\tau_k}$ and $x_{t_{k+1}}$. We then repeat this process for each step $k \in [0, N)$ to get a sample from the model. See **??** for a discussion on the performance of this sampling procedure.

