# OpenReview forum: "On Flow-based Generative Models for Probabilistic Forecasting"
_NeurIPS.cc/2025/Conference — Submitted to NeurIPS 2025_

### Official Review · Reviewer_G6pi · 2025-06-08

**Clarity:** 3
**Significance:** 3
**Originality:** 3
**Rating:** 4
**Confidence:** 3

**Summary:**

This paper proposes to use autoregressive flow based generative models for probabilistic forecasting (i.e., time series generation step by step). First, they relaxe the framework of stochastic interpolants to generalized stochastic interpolants, which do not exactly arrive at a point but at a gaussian potential. Then the authors derive several models, based on the SDEs, which have the correct path measure and also propose autoregressive solutions to solve the variational inference problem, which do not need to be trained with maximum likelihood. Experiments are conducted on several toy data sets.

**Questions:**

1) Where does the $q^{Neural_SDE}$ trained as an bayes estimator appear in the experimental tables?

2) Are the gaussian potential fixed in the experiments and how is $\sigma$ chosen then? Basically you also wrote down an ELBO/NLL so it should be okay'ish to learn EM style?

**Ethical Concerns:**

["NO or VERY MINOR ethics concerns only"]

**Final Justification:**

The authors response did briefly answer some of my questions, and others not really (eg. error accumulation). I keep my score as is, as it did not change my perception.

**Limitations:**

mainly theory paper so fine

**Quality:**

3

**Strengths And Weaknesses:**

Strengths:

1) The maths is written pretty clearly and is well done. The authors have done a great job at developing rigorous mathematics for their algorithms.

2) The idea is very timely. Whereas generative AI has made great progress on images, the generation of time series (or probabiistic forecasting) lags behind. Further, there is great application potential in finance.

Weaknesses:

1) The framework of generalized interpolants is a bit confusing to me/the motivation is not super clear. So adding a bit of Gaussian noise at the time series end points introduces an error which blurs the distribution a bit and therefore could hinder precision. What is the intuition behind WHY one should do it.

2) The model in the experiments is an LSTM type model. This is in conflict with the continuous perspective (SDE) since LSTMs have 'fixed' hidden states/discretization. If one would employ the time as a continuous input, this would be more clear. (Correct me if i am wrong here)

3) The paper does not really compare to other state of the art algos. While I mainly see this as a nice theoretical paper, I would argue that getting a feeling for how well other algos perform should be there.

One interesting comparison could be: Trajectory Flow Matching with Applications to Clinical Time Series Modeling, Zhang et al, Neurips 2024

4) The paper sometimes states that the extension to the irregular case is simple. I see this with the autoregressive models but with the FBGM i dont really understand how this should work.

---

> ### Author Rebuttal · Authors · 2025-07-30
>
> Thank you for the detailed review, we will improve our paper based on your feedback.  We used Gaussian potentials to encapsulate the setting used in traditional Gaussian time series models, such as when observations $y$ are sampled from a Gaussian emission distribution $p(y|x)$.  The RNN used in our experiments was conditioned on the time steps in order to make it work with continuous time.  Despite our efforts, we were not get $q^{Neural_SDE}$ to generate samples reliably in practice which is why it was omitted from the results table.  The variance of the Gaussian potentials was chosen by hand as a pre-processing step so that samples from the CRF, conditioned on sequences from each dataset, looked reasonable qualitatively.  You are correct in saying that we could have learned this parameter as well because the likelihood of the CRF is available in closed form, but we chose not to for simplicity.

---

> > ### Comment · Reviewer_G6pi · 2025-08-01
> > **rebuttal**
> >
> > Thanks for the rebuttal.I am keeping my score as is.

---

### Official Review · Reviewer_RmNZ · 2025-07-01

**Clarity:** 2
**Significance:** 3
**Originality:** 3
**Rating:** 2
**Confidence:** 2

**Summary:**

The paper investigates flow-based generative models for forecasting and generalizes stochastic interpolation via Gaussian potential, leading to a Gaussian random field with relaxed endpoints. The resulting model is a latent SDE via variational inference and closely relates to previous MSE-based autoregressive models. The authors demonstrate their methodology on multiple synthetic datasets using the NRMSE and NLL.

**Questions:**

- L71: Shouldn't it be conditional distribution instead of marginal?
- How is the likelihood evaluated (Table 1)?
- "[...] the uncertainty in the models only depend on the time in between observations and not the observations themselves." This seems like a limitation to me. Can you elaborate on that?


To increase the score, the authors would need to polish the paper, i.e., fix missing references, and include a broader experimental evaluation including real-world datasets and different baselines.

**Ethical Concerns:**

["NO or VERY MINOR ethics concerns only"]

**Final Justification:**

While the paper provides compelling theoretical contributions, the current state of the paper is not yet ready for publication (see weaknesses). The concerns were not addressed during the rebuttal.

**Limitations:**

While the authors discuss the connection to previous time series models, I encourage them to include an additional limitations paragraph.

**Paper Formatting Concerns:**

Multiple equations in the appendix extend the text width.

**Quality:**

2

**Strengths And Weaknesses:**

### Strengths

- The CMFVI framework generalizes stochastic interpolation, resulting in a latent SDE via variational inference. All assumptions and modelling choices are theoretically motivated and justified.
- Gaussian potential functions allow endpoint relaxations and incorporate fixed ones as special cases.
- The authors connect to stochastic interpolants, describe how their framework relates to previous models, and show that their model is similar to autoregressive MSE-based models.
- The code is attached, and the appendix contains additional information and details on the implementation.

### Weaknesses

- The paper reads like an incomplete draft. There are various typos, incorrect grammar, missing citations, missing references, and missing proofs. (see Minors for examples)
- The experimental evaluation is insufficient. Only synthetic datasets are used, and no standard baselines are included. It is hard to see a trend in these. Furthermore, there is no discussion of the results, and the section ends in the middle of a sentence (L352).
- There is no qualitative evaluation. An example forecast with additional confidence intervals would provide valuable insights.
- The specifics of different models in the experimental evaluation are not clear. A detailed description would help interpret the results.


### Minors

- Equation 2: There should be no $x_0$ in $s_t$
- L97-99: Sentence seems cut off
- L113: "understand" should be removed
- L130-131: "the the" -> "is the"
- L137: "(CITE)" instead of citations
- L177: in linear SDEs ***in*** ...
- L198: wide range ***of*** ...
- L324: though -> thought
- L328: depend -> depends
- Multiple references missing, e.g., L646
- Proof missing (L925)

---

> ### Author Rebuttal · Authors · 2025-07-30
>
> Thank you for the detailed review of our paper.  We will work on improving the writing and the evaluation in the future.  You are correct, line 71 should say conditional distribution.  The likelihood values in Table 1 were computed using the empirical marginal distribution of samples from the model.  For each sequence in the test set, we generated 32 forecasts for each model and then constructed the empirical distribution for each dimension of the forecasts at each time step.  We then computed the average negative log likelihood of the ground truth forecast at each dimension and time.  What we meant by the sentence about the uncertainty in not depending on the observations is that the covariance matrices of the potential functions in our CRFs are not dependent on observations, which means that the covariance matrices of the transitions of our MSE based models are not learned, but instead computed in closed form using Kalman smoothing.  This is in contrast to standard autoregressive Gaussian models that parameterize the covariance matrix of their transition distributions.  This is certainly a limitation of this class of models, but not to a massive extent because the model is still able to represent complex distributions due to the autoregressive parameterization of the means.

---

> ### Comment · Reviewer_RmNZ · 2025-08-01
>
> Thank you for the response. While some of my concerns have been addressed, various issues regarding clarity remain. Furthermore, the experimental evaluation remains insufficient. Therefore, I will maintain my score.

---

### Official Review · Reviewer_6acY · 2025-07-02

**Clarity:** 3
**Significance:** 4
**Originality:** 4
**Rating:** 3
**Confidence:** 3

**Summary:**

The work proposed a solution to apply flow-based generative modelling to the time series setting. They show connections of flow-based generative modelling with a mean field variational inference algorithm for conditional exponential family distributions and Markovian projection to propose a discrete time version of flow-based generative models that use Stochastic differential equations. The theoretical gains are supported by empirical evidence from experiments on a bunch of synthetic datasets.

**Questions:**

Please see the strengths and weaknesses above.

**Ethical Concerns:**

["NO or VERY MINOR ethics concerns only"]

**Final Justification:**

The paper has a limited evaluation, which has not been addressed in the current rebuttal.

**Limitations:**

I have not read any limitations addressed directly in the main paper.

**Paper Formatting Concerns:**

None!

**Quality:**

4

**Strengths And Weaknesses:**

I think the paper is overall strong with a good theoretical contribution. I will add some of the minor points I have here.

1. The paper is overall well written, but there are some lines that get overly long and require reading them multiple times to understand them, i.e., the readability can be improved at times. A good example is in the abstract, i.e "We show that FBGMs based on linear stochastic differential equations are instances of a more general mean-field variational inference algorithm for conditional exponential family distributions that constructs Bayes estimators of natural parameters."

2. Line 19: This is not inherently true for flow matching, which works with the velocity field, and sample generation is done via ODE. Maybe it’s better to write ’stochastic process’ instead of SDE.

3. Line 37-38: The Authors wrote "several years ago", but the citations are rather recent; consider rephrasing or updating to relevant citations.

4. I think generalising to a large class of models (flow-based generative models) was a good idea, but the entire analysis seems to be heavily based on Stochastic Interpolants; it would be better to make it clear early in the paper.

5. Line 98-99: The statement seems incomplete.

6. Line 5 102-104, The statement looks repeated as the one in the introduction.

7. Probably my biggest critic if any would be the limited evaluation, it would be good to have more experiments not only synthetic datasets.

---

> ### Author Rebuttal · Authors · 2025-07-30
>
> Thank you for the constructive feedback.  We will use your suggestions to improve the exposition and work on better evaluation in the future.

---

### Official Review · Reviewer_QHMT · 2025-07-03

**Clarity:** 1
**Significance:** 4
**Originality:** 3
**Rating:** 3
**Confidence:** 1

**Summary:**

This paper studies how to adapt flow-based generative models to forecasting for time series data. The authors show why directly using these models for time series is hard in practice and propose a new way to adapt them into a simpler, discrete-time version. They connect this idea to variational inference and show that their method links naturally to common time series models like mean-squared-error predictors and Gaussian autoregressive models. They test the method on synthetic data and show it keeps useful properties of flow models while being easier to train for forecasting.

**Questions:**

1. Your method uses Gaussian potential functions for the stochastic interpolation and CRF parts. Could this choice limit how well the model can handle time series with strongly non-Gaussian or multimodal behavior? Do you have ideas for extending the framework to use more flexible (non-Gaussian) potentials?
2. Can you clarify how the choice of discretization level (e.g., number of time steps or grid resolution) affects model accuracy and computational cost? Is there a practical guideline for choosing how fine or coarse the discrete time steps should be for different types of time series?

**Ethical Concerns:**

["NO or VERY MINOR ethics concerns only"]

**Final Justification:**

After the rebuttal and discussion period, in my opinion the weaknesses of the paper remained the same, i.e. need for improved clarity and lack of complex experiments. I therefore will keep my score as is.

**Limitations:**

yes

**Quality:**

2

**Strengths And Weaknesses:**

## Strengths
The paper tackles a clear gap in flow-based generative models by making them more practical for time series forecasting. It gives a clear and solid theoretical way to move from complex continuous-time models to simpler discrete-time ones that are easier to train. The main ideas are well motivated, connect nicely to known models, and could help bring modern generative tools into everyday forecasting tasks. The experiments, while simple, show that the method works as intended.
## Weaknesses
The main weakness is that the experiments are only on simple synthetic data, so it’s unclear how well the method works on real-world time series. Also, from the results in the selected benchmarks, it is unclear whether the method perform better than the baselines. Furthermore, no comparison to strong baselines is provided, making it hard to position this method in the broader literature of models for forecasting.
Some parts of the paper are quite dense, and I think there is a lot of room for improving the exposition, as it took several passes before getting a grasp of the proposed method. Some parts of the text should be polished:

- line 99 is it missing something?
- line 113 should use understand or find, not both
- line 137 missing citation?
- line 352 missing part of sentence

---

> ### Author Rebuttal · Authors · 2025-07-30
>
> Thank you for the review.  We will work on improving our paper using your feedback.  To answer your questions, our method uses Gaussian potential functions in order to perform inference in the CRFs efficiently, and so we do not envision a simple way to use non-gaussian potentials easily with this framework.  However, we note that the flexibility of the base CRF can be increased by using more advanced probabilistic models that employ auxiliary variables, such as [1].  We found that using finer discretization levels did not have an effect on the model’s forecasting performance.  Although using a finer discretization should, in theory, increase the model capacity and make the autoregressive MSE model closer in path measure to the neural SDE, we found that for the datasets that we tested that this had no impact on the final performance.  We believe that this is because conditional Gaussian distributions are sufficient for modeling the time series data we considered.
>
> [1] - Linderman, Scott, et al. "Bayesian learning and inference in recurrent switching linear dynamical systems." Artificial intelligence and statistics. PMLR, 2017.

---

> > ### Comment · Reviewer_QHMT · 2025-08-01
> >
> > I thank the reviewers for their answers. The main weaknesses of the paper, as also mentioned by other reviewers, remain the lack of clarity in some parts, and the limited experimental evaluation. I therefore confirm my score.

---

### Decision · Program_Chairs · 2025-09-17

**Decision:**

Reject

**Comment:**

The initial reviews of this paper were mixed, with ratings between 2 and 5 (average rating: 3.5). While the reviewers acknowledged the convincing motivation/positioning, strong theoretical contribution, and appealing generalization of traditional approaches, there were also significant concerns. The dense/convoluted presentation and minimal/synthetic-only experiments, in particular, were pointed out as major weaknesses. During the rebuttal phase the authors clarified some questions, e.g., related to the role of the Gaussian potentials and the discretization level. However, the lack of strong baselines and qualitative analyses, as well as the unpolished state of the manuscript, were not adequately addressed. As a result, the average rating of the paper further decreased during the internal discussions. While all reviewers agree on the appeal of the theoretical contribution, there was also consensus that the aforementioned weaknesses prevent the paper from being accepted in its current form.

I agree with this sentiment and can therefore not recommend the paper for acceptance. I encourage the authors to revise the presentation, compare the proposed approach to strong baselines on real-world data, and resubmit their work at a later time.